# Regulatory mechanisms of lipopolysaccharide synthesis in *Escherichia coli*

Sheng Shu[1] & Wei Mi [1,2] ✉

Lipopolysaccharide (LPS) is an essential glycolipid and forms a protective permeability barrier for most Gram-negative bacteria. In *E. coli*, LPS levels are under feedback control, achieved by FtsH-mediated degradation of LpxC, which catalyzes the first committed step in LPS synthesis. FtsH is a membrane-bound AAA+ protease, and its protease activity toward LpxC is regulated by essential membrane proteins LapB and YejM. However, the regulatory mechanisms are elusive. We establish an in vitro assay to analyze the kinetics of LpxC degradation and demonstrate that LapB is an adaptor protein that utilizes its transmembrane helix to interact with FtsH and its cytoplasmic domains to recruit LpxC. Our YejM/LapB complex structure reveals that YejM is an anti-adaptor protein, competing with FtsH for LapB to inhibit LpxC degradation. Structural analysis unravels that LapB and LPS have overlapping binding sites in YejM. Thus, LPS levels control formation of the YejM/LapB complex to determine LpxC protein levels.

LPS, a glycolipid located at the cell surface of Gram-negative bacteria[1], consists of lipid A, a core oligosaccharide, and a long polysaccharide (O antigen)[2]. The LPS precursor, lipid A core, is synthesized at the cytoplasmic side of the inner membrane (IM). It is then flipped by MsbA to the periplasmic leaflet[3,4], assembled with O antigen, and transported to the cell surface by the Lpt system[5,6] (Supplementary Fig. 1a provides a schematic representation of LPS synthesis, transport, and regulation). As a lipid essential for the viability of most Gram-negative bacteria, LPS synthesis is under tight control:[7–9] too little LPS compromises the outer membrane (OM), triggering cell envelope stress responses and leading to cell death; too much LPS breaks the balance between LPS and phospholipids, and accumulation of LPS in the IM is toxic and lethal. In *E. coli*, cellular levels of LPS are controlled by LpxC, a deacetylase that performs the first committed step of synthesizing LPS[10]. As the rate-limiting step in LPS synthesis, LpxC becomes a regulatory focal point, and its protein levels can change over a 20-fold range depending on the cellular LPS levels[10,11]. The LpxC protein levels are determined by its degradation rates[11], which are mainly achieved by the membrane-anchored AAA+ (ATPase associated with diverse cellular activities) protease FtsH[12,13]. FstH is the only AAA+ protease essential for *E. coli* viability because of its role in degrading LpxC and thus controlling LPS levels. It utilizes energy from ATP

hydrolysis to unfold and translocate its substrates through an axial pore into the interior catalytic sites of the protease domain[14–16] (Supplementary Fig. 1b). In addition to LpxC, FtsH substrates include the cytoplasmic proteins RpoH[17], phage protein λCII[18], and many membrane proteins[19–21]. How cellular LPS levels specifically regulate FtsH protease activity toward LpxC––without affecting the degradation of other FtsH substrates––has been a mystery for decades.

Potential players that regulate LpxC degradation in response to LPS levels have been revealed by genetic approaches recently. One is the essential membrane protein LapB (also known as YciM), which modulates LPS levels by reducing levels of cellular LpxC[22,23]. Inactivating the *lapB* gene stabilizes LpxC, resulting in an accumulation of LPS and its precursors, eventually leading to cell death[22,23]. Conversely, overexpressing LapB reduces LpxC levels significantly[23]. LapB-mediated regulation of LpxC is contingent on FtsH protease activity, and remarkably, LapB does not influence levels of FtsH substrates other than LpxC[22,23]. It was also proposed that LapB is an LPS assembly protein based on the finding that LapB interacts with many enzymes and proteins involved in LPS synthesis and transport[22].

Another key player is the essential membrane protein YejM[24], which includes a TM domain, a linker, and a periplasmic domain

---

[1]Department of Pharmacology, Yale University School of Medicine, New Haven, CT 06520, USA. [2]Department of Molecular Biophysics and Biochemistry, Yale University, New Haven, CT 06520, USA. ✉e-mail: wei.mi@yale.edu

(Supplementary Fig. 1b). YejM functions to restrain FtsH-dependent LpxC degradation[13,25–28]. Deleting the *yejM* gene promotes LpxC degradation, leading to reduced LPS levels. Genetic analysis suggested that YejM acts upstream of LapB, and that the inhibitory effect of YejM depends on active LapB[25,26]. In support of this, bacterial two-hybrid experiments, pull-down assay, and immunoprecipitation demonstrated that YejM and LapB form a complex[13,26,29]. However, there are also genetic data suggesting that YejM may bypass LapB and regulate LPS synthesis directly. In support of this, deletion of the *lapB* gene is rescued by suppressors mapping to the *yejM* gene[13], and *lapB* mutants that rescue truncations of *yejM* have a non-detectable level of LapB protein in cells[27]. As such, it was proposed that YejM acts as an antagonist of LapB[13]. Intriguingly, an LPS molecule was found binding to YejM in a recent crystal structure of YejM, suggesting that YejM is a sensor of LPS[29]. This discovery provides a glimpse of how LPS levels are sensed in the IM to regulate LpxC degradation. Despite the enormous progress in the field, the molecular mechanisms of regulation, such as how LapB stimulates LpxC degradation and how LPS regulates the antagonistic function of YejM, remain unresolved.

In this work, we use in vitro approaches to study the regulatory mechanisms of LpxC degradation with purified components. We establish an in vitro assay to study the kinetics of LpxC degradation and also determine a cryo-electron microscopy (cryoEM) structure of the YejM/LapB complex. Our data support a model in which LapB binds to FtsH and acts as an adaptor protein to stimulate LpxC degradation specifically. We also identify an 'anti-adaptor' function for YejM, which sequesters LapB under conditions of low LPS to reduce LpxC degradation by FtsH.

## Results

### LapB is an adaptor protein specific for FtsH-mediated LpxC degradation

We tested the protease activity of purified FtsH with- or without LapB on different substrates, including LpxC, RpoH, λCII, and a generic protease substrate β-casein. We monitored the degradation of substrates by assessing their disappearance on SDS-PAGE gels (Supplementary Fig. 2). Only FtsH-mediated degradation of LpxC, but not other substrates, was significantly accelerated by the addition of LapB. In controls, purified LapB alone did not degrade LpxC, excluding the possibility that other contaminating proteases contributed to accelerated LpxC degradation. These results biochemically recapitulated previous genetic data indicating that LapB can specifically stimulate FtsH proteolysis of LpxC[23].

To understand how LapB stimulates FtsH-mediated LpxC degradation, we labeled LpxC with a fluorescent dye, Atto488, and quantitated LpxC degradation rates by measuring the fluorescence signal from the soluble digestion products (Atto488-labeled peptides) after precipitating and removing undigested LpxC (Supplementary Fig. 3a). We first used SDS-PAGE to verify that Atto488 labeling did not affect LpxC degradation kinetics. Indeed, the intensities of both the stained and in-gel fluorescence bands from labeled LpxC disappeared at similar rates to unlabeled LpxC (Supplementary Fig. 3b). We then reconstituted FtsH into proteoliposomes with or without LapB and measured the increase in fluorescence signal from digested peptides (Supplementary Fig. 3c). Plotting initial reaction rates against substrate concentrations (Fig. 1a) revealed that LapB reduced the Michaelis constant ($K_M$) for LpxC by approximately eightfold, whereas there was only a 7% increase in the maximum velocity ($V_{max}$). These data suggest that LapB is an adaptor protein that enhances the binding affinity of FtsH for LpxC.

As an adaptor protein, LapB should be able to bind to FtsH. After co-expressing non-tagged FtsH and His-tagged LapB, we found that FtsH co-eluted with LapB$_{His}$ during affinity purification (Fig. 1b), suggesting that LapB and FtsH form a stable complex. LapB has a single N-terminal TM helix and large cytoplasmic domains[30,31]

(Supplementary Fig. 1b). We next asked whether the N-terminal TM helix plays an active role in interacting with FtsH or simply anchors LapB to the membrane. To distinguish between two possibilities, we selected three proteins (AcrZ, DjlA, and KdtA) that each has a single N-terminal TM helix without significant sequence similarity to the LapB TM helix. We used these different TM helices to replace the TM helix in LapB. If the LapB TM helix merely functions as an anchor and does not interact with FtsH, replacing it with another TM helix should still allow LapB anchoring to the IM and form a stable complex with FtsH. However, co-expressing and purifying the His-tagged LapB chimeras with FtsH resulted in significantly reduced levels of co-eluted FtsH (Fig. 1b). These results argue that the LapB TM helix plays an important role in specific interactions with FtsH.

We speculated that the cytoplasmic domain of LapB might bind and recruit LpxC for degradation by FtsH. To test this hypothesis, we purified the LapB cytoplasmic domain (LapB$_{cyto}$) and added it to the LpxC digestion assay with FtsH/LapB proteoliposomes. Since LapB$_{cyto}$ only has the cytoplasmic soluble domain and is not incorporated into proteoliposomes, its binding to LpxC in solution would reduce the amount of LpxC recruited to FtsH/LapB in the proteoliposomes and thus slow down LpxC degradation by FtsH/LapB proteoliposomes (Supplementary Fig. 3d). As expected, increasing LapB$_{cyto}$ levels progressively inhibited LpxC degradation (Fig. 1c). We estimated an IC$_{50}$ of 3.96 μM (2.64–6.13 μM, 95% confidence interval) based on the inhibition curve and calculated a dissociation constant ($K_d$) of 1.34 μM between LapB$_{cyto}$ and LpxC. This $K_d$ is similar to the $K_M$ value (1.93 μM) in the kinetic studies of LpxC degradation by the FtsH/LapB proteoliposomes, which suggests binding between LpxC and the cytoplasmic domain of LapB is a major contributor to the increased affinity of LpxC to the FtsH/LapB complex. In summary, LapB appears to function as an FtsH adaptor protein, with its TM helix interacting with FtsH and its cytoplasmic domain directly recruiting LpxC for degradation by FtsH.

### Structure of the YejM/LapB complex

Previous studies suggested that the inhibitory effect of YejM depends on active LapB[25] and that YejM and LapB form a complex[13,26,29]. To understand how YejM affects LapB function, we purified the YejM/LapB complex in detergent GDN and determined its structure at an estimated overall resolution of 3.9 Å using single particle cryoEM (Supplementary Figs. 4a–c & 5). The TM domains of the complex had the highest resolution at 3.3 Å, with well-defined densities for most of the sidechains (Supplementary Figs. 6a, 7). The cytoplasmic domains of LapB had a relatively lower resolution, and the periplasmic domains of YejM were not visible in this 3.9 Å map because of their flexibility. The flexibilities in YejM periplasmic domains (Supplementary Fig. 8a) are reminiscent of EptA, a lipid A phosphoethanolamine transferase sharing a similar fold with YejM[32]. To position these periplasmic domains, we carried out another round of 3D classification with local angular searches (Supplementary Fig. 5c). Particles from the class that has the strongest periplasmic densities were further refined to 4.1 Å (Supplementary Fig. 5d). The resolution of the periplasmic domains in this map is still not high enough for atomic model building (Supplementary Fig. 6b), but allowed us to unambiguously dock the crystal structure of this domain (Supplementary Fig. 8)[29]. The docked periplasmic domain is rotated ~80 degrees compared to the crystal structure of YejM and ~60 degrees relative to the EptA structure (Supplementary Fig. 8e, f) after superimposing their TM domains. In the final 3D model, we built only the TM and cytoplasmic domains of the YejM/LapB complex based on the 3.9 Å map (Supplementary Figs. 5b & 6c).

The structure revealed two LapB molecules (LapB_A and LapB_B) and two YejM molecules (YejM_C and YejM_D) in the complex (Fig. 2a, b). To rule out the oligomeric state was an artifact introduced by detergent, we purified the complex with styrene maleic acid (SMA), a polymer that solubilizes membrane proteins whilst maintaining their

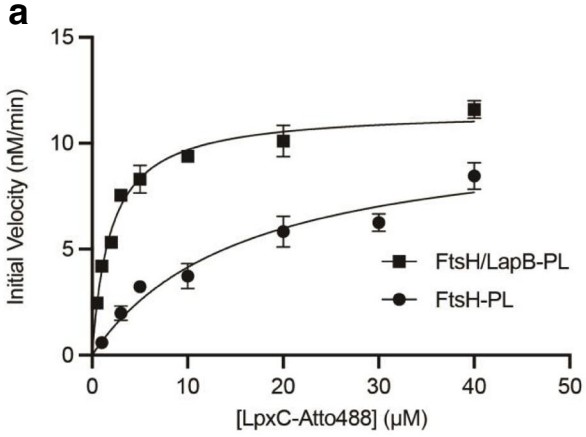

|  | FtsH-PL | FtsH/LapB-PL |
|---|---|---|
| **Best-fit values** |  |  |
| $K_M$ (µM) | 15.90 | 1.93 |
| $V_{max}$ (nM/min) | 10.74 | 11.56 |
| **95% Confidence interval (CI)** |  |  |
| $K_M$ (µM) | 9.87-27.38 | 1.62-2.29 |
| $V_{max}$ (nM/min) | 8.92-13.78 | 11.05-12.09 |
| **Goodness of fit** |  |  |
| $R^2$ | 93.35% | 96.87% |

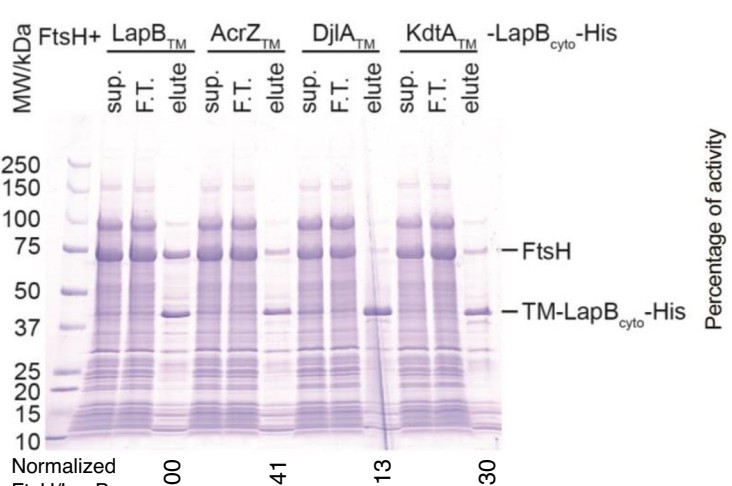

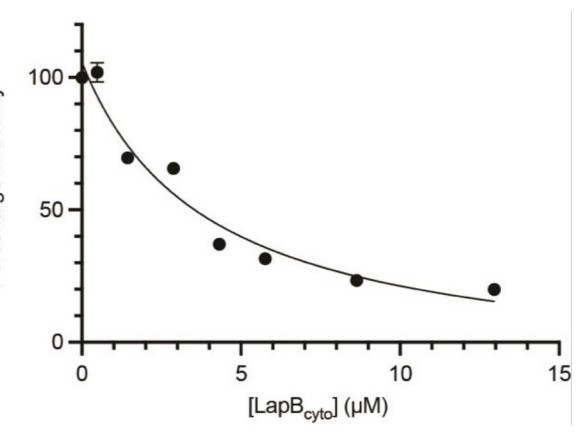

**Fig. 1 | LapB is an adaptor specific for FtsH-mediated LpxC degradation.**
**a** Kinetic analysis of LpxC degradation with proteoliposomes (PL). The initial rates of LpxC degraded by FtsH or FtsH/LapB proteoliposomes were fitted to the Michaelis-Menten equation, and $K_M$ and $V_{max}$ were estimated. **b** Effect of replacing LapB's TM helix. His-tagged wild-type or chimeric LapB proteins were expressed and affinity-purified from bacteria, and co-elution of FtsH was examined. The normalized ratio of SDS-PAGE intensities of co-eluted FtsH and LapB bands are listed below the lane of elution. Sup., supernatant of detergent-solubilized membrane after ultracentrifugation; F.T., flow-through. **c** LapB$_{cyto}$ concentration-dependent inhibition of LpxC degradation by the FtsH/LapB proteoliposomes. The degradation rate without LapB$_{cyto}$ was set to 100% and used to normalize rates of LpxC degradation with different LapB$_{cyto}$ concentrations. For the kinetic analysis and inhibition assay, each experiment was repeated three times and all data were presented as mean values with error bars representing standard deviations (SDs) of triplicates. Source data for (**a**) and (**c**) are provided as a Source Data file.

native state in phospholipid bilayer[33]. Negative stain EM analysis showed that the SMA solubilized YejM/LapB complex maintained the same oligomeric state as that purified with the detergent GDN (Supplementary Fig. 4d–f). Therefore, our cryoEM structure of the YejM/LapB complex should represent a native state in the membrane. The two LapB molecules form a homodimer with their single TM helices in the center of the TM bundles of the complex. The TM helices of the two YejM molecules are located in the periphery of the integral membrane part of the complex, sandwiching the TM helices of the LapB dimer— which mediate YejM dimerization, with no direct interactions between the two YejM molecules in the membrane. The densities of the two YejM molecules were not equally good, with YejM_C having better sidechain densities than YejM_D (Supplementary Fig. 7). The TM domains of the complex have a pseudo-two-fold symmetry perpendicular to the IM (Fig. 2c). However, the whole complex has no symmetry as the cytoplasmic domains of LapB are at different distances to the IM, with LapB_B close to and LapB_A far from the membrane (Fig. 2b).

## Lipid-mediated interactions between YejM and LapB
Because of the two-fold symmetry in the TM domains and better densities in the YejM_C molecule, we focused on the interfaces between YejM_C and the LapB dimer to visualize their interactions (Fig. 3). Protein-protein interactions between YejM_C and the LapB dimer occur largely within the membrane, mainly mediated by hydrophobic side chains (Supplementary Fig. 9). They can be divided into two regions (dashed rectangles in Fig. 3a), which align well with the two leaflets of the inner membrane (Fig. 3a). The LapB_A TM helix and the TM helices 1&2 of YejM_C interact within the cytoplasmic leaflet (Supplementary Fig. 9), leaving a large space between them in the periplasmic leaflet. On the other hand, the LapB_B TM helix interacts with YejM TM helices 2&5 in the periplasmic leaflet (Supplementary Fig. 9), leaving an intervening space in the cytoplasmic leaflet. Interestingly, in these two "empty" spaces, which are marked by grey trapezoids in Fig. 3a, we observed lipid-like densities (Fig. 3b). Although we could not unequivocally identify what lipids these represent based on the cryoEM densities, we were able to fit two

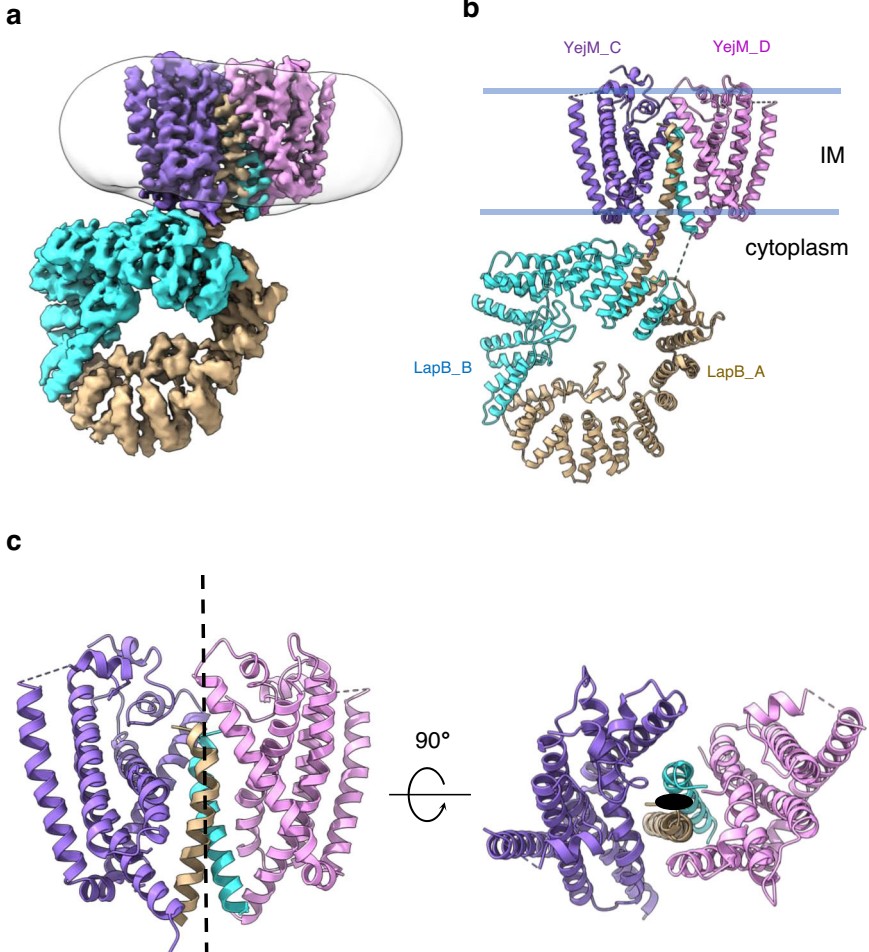

**Fig. 2 | CryoEM structure of the YejM/LapB complex. a** Surface view of 3D reconstruction of the YejM/LapB complex, filtered to 3.9 Å resolution. Volumes for proteins are presented with a contour level of 0.25. Molecules of LapB_A, LapB_B, YejM_C, and YejM_D are colored in brown, cyan, purple, and pink, respectively. The volume for the detergent micelle is smoothened with a Gaussian filter and shown as a transparent outline. **b** Ribbon diagram of an atomic model of the YejM/LapB complex, with the thick blue lines indicating the boundaries of the inner membrane (IM). **c** Side and top views of the TM domains of the YejM/LapB complex, with a pseudo-two-fold axis shown as a dashed line or a dot.

glycerophospholipid molecules into them. We modeled an unsaturated phosphatidylglycerol in the lipid-like densities at the periplasmic leaflet (Lipid$_{Peri}$), with one acyl chain folded back to the periplasmic side to interact with the N-terminus of the LapB_A TM helix (Supplementary Fig. 10a). We modeled a phosphatidic acid into the cytoplasmic leaflet density (Lipid$_{Cyto}$), whose phosphate moiety forms a salt bridge with Arg22 in LapB_A (Supplementary Fig. 10b).

### LapB and LPS bind overlapping sites in YejM

Our 3.9 Å resolution map allowed us to unambiguously model residues 2-222 of YejM_C, which includes all the TM helices (residues 1–190) and a partial linker region. We superimposed the YejM_C structure with the previously described YejM/LPS structure[29] and found a dramatic change in the orientation of the linker region (Fig. 4a). In the YejM/LPS structure, the residues 210-YPMTARRF-217 in the linker region are parallel to the membrane surface and bind to LPS[29]. By contrast, in our YejM/LapB structure, this linker is perpendicular to the membrane, protruding into the periplasmic spaces (this orientation is further validated by the 4.1 Å map, as shown in Supplementary Fig. 8c). In this context, the YejM_C conformation seen in the YejM/LapB complex cannot bind to LPS located in the IM, because the key LPS-binding residues are too distant from the IM. Intriguingly, in comparison of our YejM/LapB complex structure with the YejM/LPS crystal structure, we found that the TM helices of the LapB dimer and LPS occupy

overlapping sites in the YejM molecule (Fig. 4b, c and Supplementary Movie 1), arguing that the binding of LPS and the LapB dimer to YejM are mutually exclusive—and that they compete for YejM binding. To test this hypothesis, we incubated the purified YejM/LapB$_{His}$ complex with LPS and ran it through the affinity column. YejM was washed away, and only LapB eluted, suggesting that the complex dissociates when LPS is present. In controls that other phospholipids were incubated with the complex, YejM and LapB co-eluted after extensive washing, indicating a stable complex with lipids other than LPS (Supplementary Fig. 11).

### YejM senses LPS levels and acts as an anti-adaptor to regulate LpxC degradation

In the YejM/LapB complex structure, the two YejM molecules sequester the TM helices of the LapB dimer. Since LapB exploits its TM helix to interact with FtsH, YejM binding to LapB will prevent LapB from functioning as an adaptor protein to accelerate LpxC degradation; thus, YejM is an anti-adaptor protein. As structural analysis suggested that LPS competes with LapB for the same binding site in YejM, high LPS levels in the IM could displace LapB, which in turn would be free to bind FtsH and stimulate LpxC degradation. To test this hypothesis, we reconstituted all components of the system into proteoliposomes to recapitulate this process in vitro. As shown in Fig. 4d, proteoliposomes with FtsH/LapB had about 2.5 times higher LpxC degradation activity

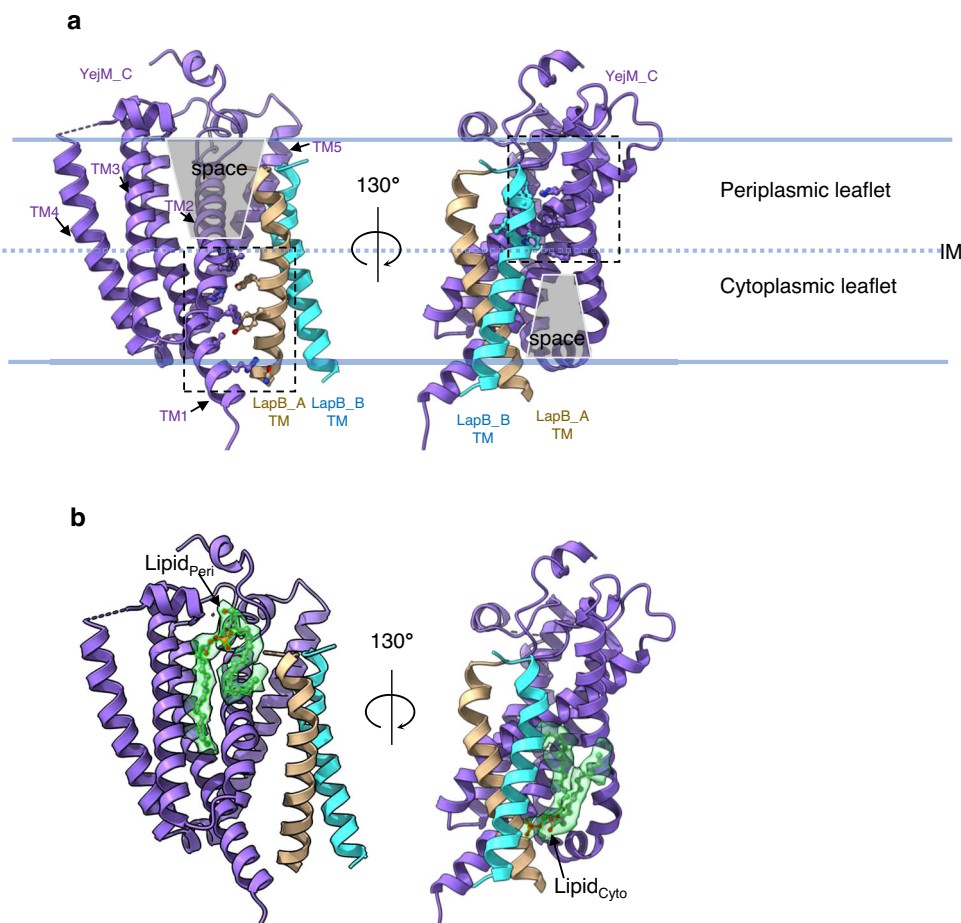

**Fig. 3 | Lipid-mediated interactions between the YejM_C and the LapB dimer.**
**a** Interactions between the YejM_C and each LapB molecule are located in the different leaflets of the IM. The thick blue lines indicate the boundaries of the inner membrane (IM) and the dashed line indicates the interface of two leaflets. Dashed rectangles highlight protein/protein interactions in each leaflet. Residues involved in interactions within 3.5 Å are shown as balls and sticks. The "empty" spaces at the interfaces are shaded with grey trapezoids. **b** Lipids at the interface between YejM_C and the LapB dimer. Atomic models of lipid molecules are shown as sticks and balls within their cryoEM densities (green, presented with a contour level of 0.25).

than FtsH proteoliposomes, whereas FtsH/LapB/YejM proteoliposomes had similar activity to that of FtsH proteoliposomes – indicating that YejM acts on LapB to exert its inhibitory effect. In proteoliposomes containing Kdo$_2$-Lipid A (a precursor of LPS), inhibition by YejM was partially released (Fig. 4d). The effect of Kdo$_2$-Lipid A is dose-dependent: the inhibitory effect of YejM was better released by a higher LPS concentration (Supplementary Fig. 12). In proteoliposomes with cardiolipin as a control, YejM retains its inhibitory ability. Previous studies suggested that introducing negatively-charged residues in the linker region abolishes LPS binding to YejM[29,34]. Consistently, a YejM mutant, YejM$^{3D}$ with three mutations (T213D/R215D/R216D), still inhibited LpxC degradation, but the inhibitory effect of this YejM mutant was not reversed by Kdo$_2$-Lipid A (Fig. 4d). Thus, the biochemical assay further supports our claim that YejM senses LPS in the IM and acts as an anti-adaptor to regulate LpxC degradation.

## Discussion

Our in vitro degradation assays supported that LapB is an adaptor protein specific for FtsH-mediated LpxC degradation. Although originally named "lipopolysaccharide assembly protein B"[22], our data suggest that LapB is best thought of as "LpxC degradation adaptor protein B". By contrast, our YejM/LapB complex structure further suggests that YejM is an anti-adaptor protein, using its TM domain to sequester the two TM helices of the LapB dimer. This finding explains previous reports that only the TM domain in YejM is essential for *E. coli*

viability[24] and that the YejM TM domain alone is enough to maintain the LpxC levels during exponential growth[26,34].

It was suggested that the YejM/LapB complex was constitutive[29]. However, a comparison of our YejM/LapB structure with the YejM/LPS crystal structure suggests that YejM and LapB dissociate when excess LPS accumulates in the IM, as LPS and LapB have overlapping binding sites in YejM (Fig. 4b, c) and the purified YejM/LapB complex dissociated when LPS is present (Supplementary Fig. 11). Further in vivo experiments are required to investigate the dynamics of the YejM/LapB complex and clarify whether the YejM/LapB complex is constitutive or transient in their native membrane environments. The interactions between YejM and LapB are unusual: YejM directly interacts with half of a TM helix from each LapB molecule in each leaflet of the membrane, leaving the other half available for interactions with lipids. This interaction mode has an obvious advantage: the protein-protein interactions provide specificity for complex formation, but the lipid-mediated interactions prevent high-affinity association, allowing the binding between YejM and LapB to be readily reversible. These lipid-mediated interactions suggest a mechanism of reversible membrane protein complex formation, which may be an essential property for rapidly regulating LpxC degradation according to the needs for LPS in cells.

It has been known for decades that LPS and phospholipid synthesis share a common substrate and are strictly coupled by synchronization of the activities of two key enzymes, LpxC and FabZ, in the two

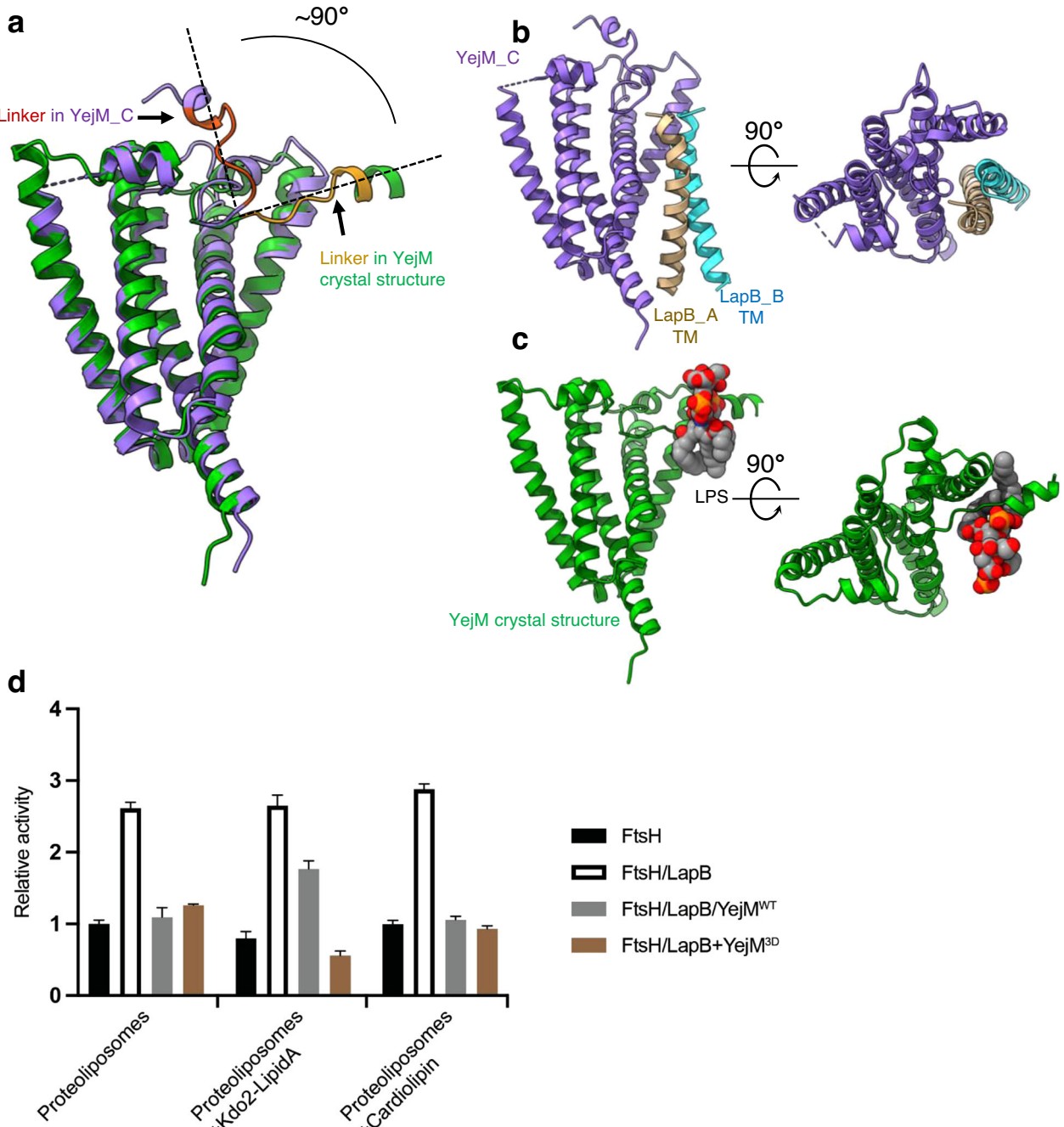

**Fig. 4 | Structural comparison of the YejM/LapB and YejM/LPS complexes and protease assays with reconstituted proteoliposomes. a** Structural comparison of the YejM_C molecule (purple, residues 210-YPMTARRF-217 in the linker region colored in orange) and the crystal structure of YejM (green, residues 210-YPMTARRF-217 in the linker region colored in dark yellow) (PDB ID: 6XLP), by superimposing the TM domains with an RMSD of 1.22 Å in the main chains. The orientations of the linker in the two structures are highlighted with dashed lines. **b** Side and top views of the TM domains of the YejM_C and the LapB dimer. **c** Side

and top views of the TM domain of YejM with an LPS molecule (shown as spheres) in the crystal structure. **d** Protease activity on LpxC of proteoliposomes reconstituted with FtsH, FtsH/LapB, FtsH/LapB/YejM$^{WT}$, or FtsH/LapB/YejM$^{3D}$ (with mutations of T213D/R215D/R216D). The activity of FtsH-proteoliposomes is set to 1, and the activities of all the other proteoliposomes are normalized to it. Each experiment was repeated three times and all data were presented as mean values with error bars representing standard deviations (SDs) of triplicates. Source data for (**d**) are provided as a Source Data file.

respective pathways[12,35,36] (Supplementary Fig. 13). Changes in FabZ activity affect FtsH-mediated LpxC degradation rates, but how this is achieved is not clear[12]. It is believed that certain intermediate metabolites in the two pathways act as signaling molecules to regulate FtsH activity towards LpxC, but the identities of the signaling molecules and their sensing receptors are under dispute. In our cryoEM map, the lipid$_{cyto}$ has no densities for a large head group, and its small head group is close to Arg22 in LapB. As such, we tentatively build a

phosphatidic acid (PA) molecule. PA is a common precursor for synthesizing all glycerophospholipids in *E. coli*, and the abundance of PA is low[37,38]. We speculate that fluctuations in PA levels may reflect the status of phospholipid synthesis, and act as signals to regulate the YejM/LapB complex formation, thus adjusting LPS synthesis accordingly. It was also suggested that the cellular level of unsaturated phospholipid might play a key role in coupling the two pathways[39]. In our YejM/LapB complex structure, the lipid$_{Peri}$ has a bent acyl chain,

suggesting that it is an unsaturated phospholipid. Interestingly, in the YejM/LPS structure, lipid molecules (only one acyl chain was modeled) bind to the same sites as the two phospholipids observed in the YejM/LapB complex structure (Supplementary Fig. 14), suggesting that YejM has an intrinsic affinity for phospholipids at these two sites. So, besides sensing the LPS level in the IM, we surmise that YejM may be a central node for sensing phospholipid composition in the IM and regulating LpxC degradation--thus contributing to the coupling of LPS and PL synthesis. Identification of the phospholipid molecules at the YejM/LapB interfaces will be a key step in further exploring this direction.

It was believed that YejM had a rigid periplasmic domain as this region adopted the same conformation in crystal structures determined by two independent groups[29,40]. Our 3D classification and the refined cryoEM map provide the evidence that YejM has a flexible periplasmic domain, as seen in EptA that requires this flexibility to fulfill its transferase activity. Do the flexibilities of the periplasmic domain in YejM play a role in regulating LPS synthesis? In the YejM/LapB and YejM/LPS structures, the linker region of YejM can be either lifted out of or in parallel with the membrane (Fig. 4a). A possible scenario is that conformational changes of the periplasmic domain, such as rotations of the periplasmic domain relative to the TM domain, may subsequently change the orientation of the linker region, which in turn regulates the complex formation/dissociation and LPS synthesis. Another scenario is also possible--the flexibilities of the periplasmic domain are the results of conformational changes in the linker region when it lifts out or inserts into the membrane. In this scenario, the energy required for lifting the linker region out of the IM may be compensated by the binding between YejM and LapB.

Although our YejM/LapB complex structure strongly supports that YejM acts upstream of LapB as an anti-adaptor, we cannot exclude that YejM has a direct effect on FtsH, which was suggested by genetic data[13,27]. Interestingly, we observed that YejM could slightly inhibit FtsH protease activity in proteoliposomes without LapB (Supplementary Fig. 15), suggesting that YejM may perform the antagonist role through multiple approaches. With our current structural data, we also cannot explain why deletion of YejM's periplasmic domain causes reducing levels of LPS and this effect might be the result from YejM antagonizing through approaches other than sequestering LapB from FtsH.

Based on our structural/biochemical results and previous studies in the field[25,26,29,41], we propose the following model to explain how LPS levels are regulated during *E. coli* growth (Fig. 5). When *E. coli* cells are in their exponential growth phase, OM biogenesis requires a large amount of LPS. All newly synthesized LPS molecules are transported to the OM, and very little LPS stays in the IM. In this situation, YejM and LapB form a complex, and FtsH without the LapB adaptor degrades LpxC slowly. LpxC is thus maintained at a high level, supporting the high LPS demand during robust cell growth. When *E. coli* cells enter the stationary phase, OM biogenesis slows, and there is little need for additional LPS. Newly synthesized LPS cannot be incorporated into the OM efficiently and may accumulate in the IM. An LPS buildup in the IM ultimately competes with LapB for YejM, releasing LapB from YejM. The liberated LapB binds to FtsH, and the formation of the FtsH/LapB complex accelerates LpxC degradation. LPS biosynthesis thus slows down because LpxC levels fall, bringing a new balance with the reduced needs for LPS. This simplified model may provide a basic framework that allows us to further probe the complexity of regulation of LPS synthesis.

## Methods
### Molecular cloning, protein expression, and purification
All genes or gene truncations (except for λCII, which was synthesized by Quintara Biosciences) were amplified from *Escherichia coli* strain CFT073 genomic DNA. For the chimeric LapB, the DNA sequence of transmembrane regions of AcrZ (residues 1-28), DjlA (residues 1-23), or

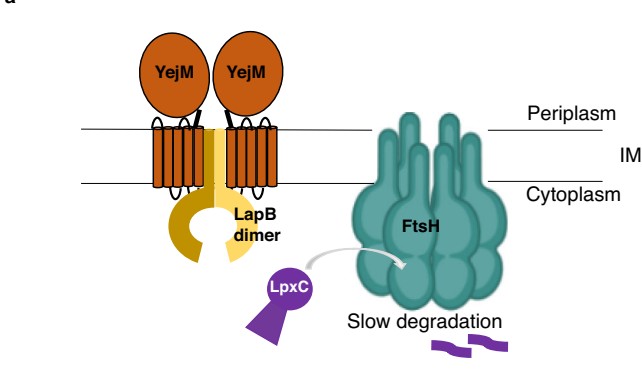

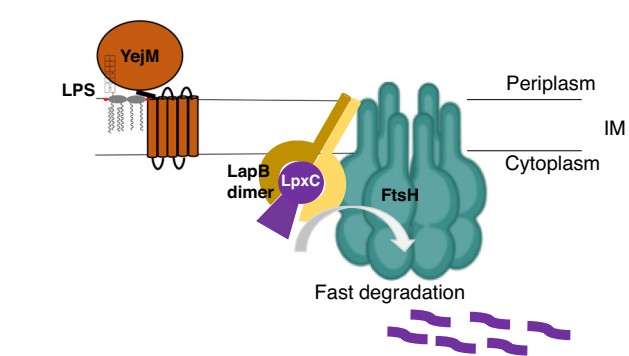

**Fig. 5 | Model for regulation of LpxC degradation. a** In exponential growth phase, YejM and LapB form a stable complex, and FtsH degrades LpxC slowly. **b** In stationary phase, LPS accumulates in the IM, displacing the LapB dimer from YejM. The liberated LapB dimer binds to FtsH, acts as an adaptor protein, and stimulates LpxC degradation.

KdtA (residues 1-29) were fused to the cytoplasmic region of LapB (aa 21-389) by overlap PCR. Sequences of primers are listed in Supplementary Table 1. The gene or gene truncations were then cloned to linearized vector using Gibson Assembly kit (New England BioLabs). All the constructs were confirmed by DNA sequencing from the Keck DNA sequencing facility at Yale University.

Overexpression and purification of FtsH: C-terminally His-tagged FtsH in pETduet vector was transformed into *E. coli* strain BL21 Star™ (DE3) pLysS. Cells were grown at 37 °C in Terrific broth (TB) medium with 50 μg/ml ampicillin and 12.5 μg/ml chloramphenicol. After the cells were grown to OD600 of ~0.6, the temperature was shifted to 30 °C, and the protein expression was induced with 0.2 mM isopropyl β-D-1-thiogalactopyranoside (IPTG) for 4 h. The cells were harvested by centrifugation and resuspended in buffer containing 50 mM Tris, pH 7.8, 300 mM NaCl, 5 mM 2-mechaptalethanol and 10% (v/v) glycerol. Cells were broken using a high-pressure microfluidizer (Emulsiflex-C3, Avestin). Cell membranes were spun down by ultracentrifugation at 40,000 rpm (185677 g) for 2 h at 4 °C using a Type 45 Ti rotor (Beckman). Membrane proteins were solubilized with 0.5 % (w/v) Lauryl Maltose Neopentyl Glycol (LMNG, Anatrace) for 1 h at 4 °C. FtsH$_{His}$ was affinity purified by TALON metal affinity resin (Clontech). For the expression of protein complexes FtsH$_{His}$/LapB$_{His}$, FtsH/LapB$_{His}$, FtsH/AcrZ$_{TM}$-LapB$_{cyto}$-His, FtsH/DjlA$_{TM}$-LapB$_{cyto}$-His, and FtsH/KdtA$_{TM}$-LapB$_{cyto}$-His, FtsH or FtsH$_{His}$ in pETduet vector and LapB$_{His}$ or chimeric LapB$_{His}$ in pRSF22b vector were co-transformed into *E. coli* strain BL21 Star™ (DE3) pLysS. Cells were grown at 37 °C in TB medium with 50 μg/ml ampicillin, 25 μg/ml kanamycin, and 12.5 μg/ml chloramphenicol. The protein expression and purification were essentially the same as that of FtsH$_{His}$

described above. LapB$_{His}$ was purified as described for FtsH purification, except that 1% (w/v) GDN (Anatrace) was used to solubilize proteins from the cell membrane.

LpxC in pETduet vector with an N-terminal His tag and 3 C cleavage site was transformed into *E. coli* strain BL21 Star™ (DE3) pLysS. Cells were grown in Lurial broth (LB) medium with 50 µg/ml ampicillin and 12.5 µg/ml chloramphenicol. After the cells were grown to OD600 of ~0.6, the temperature was shifted to 30 °C, and the protein expression was induced with 0.2 mM IPTG for 4 h. The cells were harvested with centrifugation, resuspended in buffer containing 50 mM Tris, pH 7.8, 300 mM NaCl, 1 mM TECP, and 10% (v/v) glycerol, and lysed using a high-pressure microfluidizer. After centrifugal clarification, LpxC in the supernatant was affinity purified with Ni-NTA agarose (Qiagen). The N-terminal His tag was cleaved by 3 C protease during dialysis in buffer containing 25 mM Tris, pH7.8, 150 mM NaCl and 5% glycerol. The untagged LpxC was purified by flowing through Ni-NTA agarose. LpxC was further purified by gel filtration. RpoH was expressed in BL21 Star™ (DE3) at 18 °C overnight, λCII was expressed in BL21 Gold (DE3) at 37 °C for 3 h, and LapB$_{cyto}$ was expressed in BL21 Star™ (DE3) pLysS at 30 °C for 4 h. RpoH was purified as described for LpxC purification. λCII and LapB$_{cyto}$ were purified as described for LpxC purification, except that the His tags were not removed.

For the expression of YejM/LapB complex, YejM in pCDFduet vector and C-terminally tagged LapB in pRSF22b vector were co-transformed into *E. coli* strain BL21 Star™ (DE3) pLysS. Cells were grown at 37 °C in TB medium with 25 µg/ml spectinomycin, 25 µg/ml kanamycin, and 12.5 µg/ml chloramphenicol. After the cells were grown to OD600 of ~0.6, the temperature was shifted to 30 °C, and the protein expression was induced with 0.2 mM IPTG for 4 h. The complex was purified as described for FtsH$_{His}$ purification, except that 1% (w/v) GDN (Anatrace) was used to solubilize proteins from the cell membrane. After purified by TALON metal affinity resin, YejM/LapB was applied to size-exclusion chromatography on a Superose 6 column in buffer containing 25 mM Tris, pH7.8, 150 mM NaCl, 1 mM TCEP, 2.5% glycerol and 0.01% GDN. The purity of YejM/LapB complex was judged by SDS-PAGE, and concentrated to ~15 mg/ml, and stored at −80 °C.

To purify YejM/LapB complex using SMA, inner membranes were separated (see *Separating inner and outer membranes of E. coli* below) and resuspended in buffer containing 50 mM Tris, pH 7.8, 300 mM NaCl, 1 mM TCEP, and 10% (v/v) glycerol to the final concentration of 100 mg/ml (w/v, wet weight). The inner membranes were then mixed with 0.5% SMA (XIRAN SL30010-P20) and incubated for 2 h at room temperature with gentle agitation. The insoluble fraction was removed by centrifugation at 4 °C for 30 min at 40,000 rpm and then the supernatant was incubated with TALON metal affinity resin overnight at 4 °C. The resin was washed with 50 column volumes of buffer containing 50 mM Tris, pH 7.8, 300 mM NaCl, 1 mM TCEP 10% (v/v) glycerol, and 10 mM imidazole. The YejM/LapB complex was then eluted with the same buffer containing 200 mM imidazole.

YejM: C-terminally His-tagged wildtype YejM or the T213DR215DR216D YejM mutant in pCDFduet vector was transformed into *E. coli* strain BL21 Gold (DE3). Cells were grown at 37 °C in TB medium with 50 µg/ml spectinomycin. After the cells were grown to OD600 of ~0.6, the temperature was shifted to 18 °C, and the protein expression was induced with 0.2 mM IPTG overnight. YejM was purified as described for FtsH purification, except that 1% (w/v) n-Dodecyl-β-D-Maltopyranoside (DDM, Anatrace) was used to solubilize proteins from the cell membrane for 2 h at 4 °C.

All constructs, overexpression conditions, and purification procedures of proteins or protein complexes were listed in Supplementary Table 2.

## Reconstitution of proteins into proteoliposomes
POPC (1-palmitoyl-2-oleoyl-sn-glycero-3-phosphocholine) purchased from Avanti Polar Lipids was solubilized in chloroform, dried under

nitrogen to form a lipid film, and stored under vacuum overnight. The lipid film was resuspended at a concentration of 100 mM in buffer containing 20 mM Tris-HCl, pH 7.5, 100 mM NaCl. POPC with 1.5% cardiolipin was made by mixing 1.5 mg *E. coli* cardiolipin (Avanti Polar Lipids) in chloroform with 100 mg POPC in chloroform. To make proteoliposomes of POPC with 1.5% Kdo2-Lipid A, Kdo2-Lipid A (Sigma) was resuspended in chloroform. After sonication, 1 mg Kdo2-Lipid A in chloroform was mixed with 66.7 mg of POPC in chloroform. Both POPC with 1.5% Cardiolipin and POPC with 1.5% Kdo2-Lipid A were dried and resuspended in the buffer as described for POPC liposomes. Resuspended liposomes were frozen and thawed for three times, sonicated with Branson 1200 Ultrasonic Cleaner for 10 min, then passed through a mini-extruder (Avanti Polar Lipids) with 200 nm pore 20 times, and swelled with 10% Triton-X100 at room temperature for 1 h. Swelled liposomes were then mixed with FtsH, or FtsH/YiM, or FtsH/LapB/YejM or FtsH/YejM at a ratio of 86:1 to 200:1 (w/w) for 1 h at room temperature. To remove the detergents and to incorporate proteins into the liposomes, the solution was incubated with 40 mg fresh Bio-Beads for 4 times: 0.5 h at room temperature; 1 h at room temperature; overnight at 4 °C and 1 h at 4 °C. The proteoliposomes were collected by centrifugation for 45 min at 60,000 rpm (218144 g) in an MLA-80 rotor and resuspended at ~0.45 mg/ml in buffer containing 25 mM Tris-HCl, pH 7.5, 100 mM NaCl, 1 mM TCEP.

## Protease assay
For the protease assays with unlabeled LpxC, RpoH, λCII and β-Casein as substrates, 0.24 µM of FtsH (hexamer) or FtsH/LapB complex were incubated with 0.5 µg of substrates in a 10 µl reaction system containing 50 mM Tris-acetate, pH 8.0, 10 mM Mg-acetate, 25 µM Zn-acetate, 80 mM NaCl, 100 ng/µl BSA and 1.4 mM 2-mercaptoethanol with or without 5 mM ATP at 42 °C for 1 h. The reactions were stopped by adding SDS-PAGE loading buffer and separated in 4-20% SDS-PAGE gel and stained with Coomassie blue.

For the protease assay with Atto488 labeled LpxC as substrate, purified LpxC was first applied to gel filtration to change the Tris buffer to 50 mM phosphate buffer, pH 7.6, 150 mM NaCl and concentrated to 10 mg/ml. LpxC was then labeled with Atto488 protein labeling kit (Sigma-Aldrich cat# 38371) with protein to dye ratio of 6:1 (molar ratio) following the kit manual. To measure kinetics, 0.06 µM of FtsH (hexamer) or FtsH/LapB complex on POPC liposomes were incubated with 1, 3, 5, 10, 20, 30, 40 µM or 0.5, 1, 2, 3, 5, 10, 20, 40 µM LpxC-Atto488, respectively, in 20 µl reaction system containing 50 mM Tris-acetate, pH 8.0, 10 mM Mg-acetate, 5 mM ATP, 25 µM Zn-acetate, 80 mM NaCl and 1.4 mM 2-mercaptoethanol at 37 °C. For each concentration of LpxC-Atto488, triplicates of measurements were taken, and backgrounds without adding enzyme were measured. Reaction rates were measured every 10 min for 20 min. 48 µl of 5% trichloroacetic acid was added and incubated at 37 °C for 30 min to precipitate undigested substrates as previously reported[42]. Proteins were precipitated at 14000 rpm (18800 g) for 20 min, and 50 µl from the supernatant was mixed with 60 µl of 500 mM Tris-HCl pH 8.8 and 100 µl were transferred to a 96 well assay plate (Costar REF# 3915). Fluorescence of excitation of 498 nm and emission of 520 nm was measured with a plate reader Tecan Infinite M1000 PRO. To calculate the initial velocity ($v_0$) of LpxC-Atto488 degradation, a standard curve was generated by measuring the fluorescence of a series Atto488 dilutions. The $v_0$ of LpxC-Atto488 degradation was calculated based on the dye to protein ratio, standard curve, and dilution fold during the measurements. Michaelis-Menten curves were fitted, the $K_M$ and $V_{max}$ values were estimated with GraphPad Prism 9.3.0.

To measure FtsH/LapB protease activity inhibition by LapB$_{cyto}$, 0.06 µM of FtsH/LapB complex on POPC liposomes were incubated with 1.8 µM of LpxC-Atto488 in 20 µl reaction buffer containing 0, 0.48, 1.44, 2.88, 4.32, 5.76, 8.64, 12.96 µM of LapB$_{cyto}$ respectively. Triplicates of measurements were taken at 37 °C for 20 min. The

inhibition curve was fitted and $IC_{50}$ was estimated with GraphPad Prism 9.3.0. The dissociation constant $K_d$ was calculated by the equation $K_d = (IC_{50} - 2K_M/3)/2$.

To measure LpxC-Atto488 degradation by FtsH, FtsH/LapB, or FtsH/LapB/YejM in POPC, POPC + Kdo2-Lipid A, and POPC + Cardiolipin liposomes, proteoliposomes containing 0.06 µM of FtsH (hexamer) were mixed with 3 µM of LpxC-Atto488 in 20 µl reaction buffer. Triplicates of measurements were taken at 37 °C for 20 min as described before.

### Separating inner and outer membranes of *E. coli*
*E. coli* cells were broken using a high-pressure microfluidizer. Whole cell membranes were spun down by ultracentrifugation at 40000 rpm (185677 g) for 2 h at 4 °C using a Type 45 Ti rotor and resuspended in 1 mM Tris-HCl, pH 7.5, 20% sucrose. In centrifuge tubes (Beckman recorder No. 344058), from the bottom to top, layer 14 ml of 1 mM Tris-HCl, pH 7.5, 73% sucrose, 14 ml of 1 mM Tris, pH 7.5, 53% sucrose and 9 ml of whole cell membranes in 1 mM Tris-HCl, pH 7.5, 20% sucrose. Inner and outer membranes were separated by ultracentrifugation with no braking at 23,000 rpm (95219 g) for 18 h at 4 °C using a SW28 rotor (Beckman). The inner membranes located between 20% and 53% sucrose were taken out with a pipet.

### Pull-down assay
10 µg YejM/LapB$_{His}$ complex was diluted in 200 µl of buffer containing 25 mM Tris-HCl, pH 7.8, 150 mM NaCl, 5% glycerol, 0.05% LDAO with (1 mg/ml LPS, or *E. coli* cardiolipin, or POPC) or without lipids. The samples were incubated on ice for 30 min, then mixed with 25 µl of TALON metal affinity resin and incubated for another 30 min at 4 °C with gentle agitation. The resin was washed three times with buffer containing 25 mM Tris, pH 7.8, 150 mM NaCl, 5% (v/v) glycerol, 0.005% GDN, and 10 mM imidazole. The proteins binding on beads were then eluted with the same buffer containing 200 mM imidazole. Both input and pull-down samples were separated in 4-20% SDS-PAGE gradient gel and visualized by Coomassie brilliant blue staining.

### Negative stain electron microscopy analysis
3 µL of YejM/LapB fractions eluted from TALON metal affinity resin was loaded on a carbon-coated 300 mesh copper grid (Electron Microscopy Sciences) previously glow-discharged at 25 mA for 45 s. The liquid drop was absorbed with filter paper after 40 s and quickly washed twice with a drop of 1.5% uranyl formate that was again blotted with filter paper. Finally, put the grid on drop 3 of uranyl formate, knead for 30 to 60 s, and absorbed with a filter paper. Grids were imaged using a Tecnai T12 microscope at a magnification of ×67,000. Particles were manually picked with SAMViewer and 2D classifications were generated with SAMEUL scripts[3].

### Cryo-electron microscopy data acquisition
For cryo-EM, 2.5 µl of purified YejM/LapB complex at a concentration of 6−7 mg/ml were applied to a glow-discharged C-flat™ Holey Carbon Grids (1.2/1.3, 300 mesh). Grids were blotted for 2.5 s with ~90% humidity and plunge-frozen in liquid ethane using a Cryoplunge 3 System (Gatan). Cryo-EM data were collected on a Titan Krios (FEI) at Yale west campus cryoEM facility, equipped with a K3 direct electron detector (Gatan) and an energy filter slit width of 20 eV. All cryo-EM movies were recorded in super-resolution counting mode using SerialEM[43]. Specifically, images were acquired at a nominal magnification of 64,000x, corresponding to a calibrated pixel size of 1.346 Å on the specimen level and 0.673 Å for super-resolution images. The dose rate was set to be 10.8 electrons per physical pixel per second. The total exposure time of each movie was 8.2 s, leading to a total accumulated dose of 50 electrons per Å$^2$, fractionated into 82 frames. All movies were recorded using a defocus ranging from −1.0 to −2.5 µm.

### CryoEM image processing
Dose-fractionated super-resolution movies collected using the K3 direct electron detector were binned over 2 × 2 pixels, yielding a pixel size of 1.346 Å. Motion correction, CTF defocus determination, and particle picking were carried out in cryoSPARC[44] v3.2.0 with default parameters. After 2D classification cleaning, we used pyem[45] to export selected particles to RELION[46] v3.1.2 for 3D classifications. Particles from selected classes of 3D classification were exported to cryoSPARC for non-uniform refinement[47]. All refinements followed the gold-standard procedure, in which two half data sets are refined independently. The overall resolutions were estimated based on the gold-standard Fourier shell correlation (FSC) = 0.143 criteria. Local resolution variations were estimated from two half data maps under cryoSPARC. The amplitude information of the final maps was corrected by applying a negative B-factor automatically determined by cryoSPARC, that is, B-factors of −216 Å$^2$ for the 3.9 Å map and −167 Å$^2$ for the 4.1 Å map.

### Model building and refinement
The crystal structures of LapB with PDB ID 4ZLH and YejM with PDB ID 6XLP were used as starting templates for our model building. Crystal structures of LapB cytoplasmic domain and YejM TM domain were fitted into the cryoEM map in UCSF Chimera[48]. The TM helix in LapB was predicted with AlphaFold[49] and fitted into the cryoEM map as well. The fitted model was refined in phenix.real_space_refine[50] with secondary structure restraints enabled, and the refined model was further manually adjusted in Coot[51]. Residues with well-defined sidechain densities were kept in the final model, while all other residues were mutated to alanine with CHAINSAW[52] in CCP4[53]. Due to the limited resolution of the map, the precise identities of individual lipids could not be determined. As explained in the main text, the lipid models, including 1-Vaccenoyl-2-palmitoyl-sn-glycero-3-phosphoglycerol (PGV) from PDB 3RLF and dioleoyl-phosphatidic acid (LPP) from PDB 4MX7 were fitted into the EM density and manually refined in Coot. Figures of structural analysis were prepared using UCSF Chimera and ChimeraX[48]. Structural and refinement statistics are listed in Supplementary Table 3.

### Reporting summary
Further information on research design is available in the Nature Research Reporting Summary linked to this article.

## Data availability
The three-dimensional cryo-EM density maps of the LapB/YejM complex have been deposited in the Electron Microscopy Data Bank under accession numbers EMD-25713 (the 3.9 Å map), EMD-25731 (the 4.1 Å map). Atomic coordinates for the atomic models have been deposited in the Protein Data Bank under accession number 7T6D. Source data are provided with this paper.

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

## Acknowledgements

We thank EM managers, Drs. Shenping Wu, Marc C Llaguno, and Jian-feng Lin, for their assistance during our cryoEM sample screening and data collection. The Glacios microscope that was used for screening samples was supported by an NIH S10 grant (S10OD023603). W.M. is supported by the NIH grant R01GM137068. We thank Drs. MA Lemmon, FJ Sigworth, M Hochstrasser, J Liu, TA Rapoport, M Liao, and Y Li for their critical reading and comments.

## Author contributions

W.M. conceived the project. S.H. performed all experiments, including building constructs, purifying proteins and proteoliposome reconstitution, setting up the in vitro degradation assays, and cryoEM data collection. W.M. and S.H. analyzed the data and wrote the manuscript.

## Competing interests

The authors declare no competing interests.
