## [Peer Review File · Nature Communications]

Regulatory Mechanisms of Lipopolysaccharide Synthesis in Escherichia coliREVIEWER COMMENTS

Reviewer #1 (Remarks to the Author):

The manuscript by Shu and Mi described the cryo-EM structure of YejM/LapB complex and its regulatory role in LPS synthesis through in vitro biochemical assays. The authors found that LapB is an adapter protein for FtsH mediated degradation of LpxC, an essential enzyme in LPS biosynthesis. They further discovered that a linker region of YejM, which had been found to bind LPS in a previous crystal structure, changed the conformation upon binding to LapB, which would not allow LPS binding. Thus, the authors proposed that YejM could sense the LPS concentration in the inner membrane and sequester LapB (forming the complex) when LPS concentration is low, which then prevents the degradation of LpxC and encourages the biosynthesis of LPS (a feedback loop). LPS is involved in the immune response of many pathogenic gram-negative bacteria. Therefore, it is very important to elucidate the structural and regulatory mechanism involved in the biosynthesis of LPS. This paper revealed the molecular basis underlying one of the regulatory circuits discovered from genetic studies. The assays developed in this work could be used for future studies in the field.

There are no outstanding concerns. I will support publication if the authors could address my questions and comments:

1. Does LapB only interact with FtsH through the TM domains? Are there any binding between the cytosolic domains of LapB and FtsH? It seems from affinity purification in Fig. 1b that other transmembrane helices (AcrZ, KdtA) also interact with FtsH although with weaker affinities. Are there any similarities in the TM sequences?
2. What is the stoichiometry between FtsH and LapB? Could it be measured? From the cryo-EM structure, LapB forms a dimer when bound to YejM. Does LapB also form a dimer when functioning as an adaptor?
3. Why FtsH did not degrade RpoH, CII, and Casein in Extended Data Fig. 2 (minimal activity was observed for LpxC but not the others in the absence of LapB)? A positive control would be better.
4. In the LpxC degradation assay (Extended Data Fig. 3a), does TCA also precipitate part of the digested peptides? Would a non-specific protease (such as proteinase K) serve as a control to calibrate the effect of TCA?

5. How does LapB interact with LpxC? Why LapBcyto could inhibit the protease activity? Does that mean the interactions through TMs of LapB and FtsH are necessary for the activity? The authors should provide some explanations/discussions. This is related to question 1 and 2.

6. Some 2D class averages showed both copies of YejM periplasmic domain (row1, column3 and row2, column3), why they were missed in the 3D reconstruction? Did the authors try focused classification with masks around the periplasmic domains?

7. Did the authors try mutating the LPS binding linker in YejM? Does that affect the binding of LapB?

8. The statistics of cryo-EM model looks fine. A model vs. map FSC is missing. Also it is necessary to show the threshold/contour level/rmsd for all the maps displayed.

9. Some words were misused.

e.g., in the abstract "...demonstrating that lapB is an adapter protein, which explores its transmembrane helix to interact with..."

Page 4, last paragraph, "Since LapB exploits its TM helix to interact with..."

Reviewer #2 (Remarks to the Author):

>very noteworthy results

>work will be a huge impact on the field and significantly advance the understanding of LPS synthesis/regulation and feedback

>conclusions and claims are generally well supported by the data

>some revision / clarification is needed, as exemplified below

>solid methodology and standards met

>sufficient detail in methods provided

Multi-drug resistant Gram-negative bacterial (GNB) pathogens represent an urgent global threat. The outer membrane (OM) of GNB is essential for life and virulence and forms a major barrier to most antibiotic discovery efforts. The unique lipopolysaccharide (LPS) molecule found in most GNB is a key molecule within the OM, and its synthesis is tightly regulated through bacterial growth, but the underlying mechanisms of this regulation have remained enigmatic for >30 years. The present study by Shu and Mi sheds key and new light on the molecular and structural details of LPS regulation, and is truly an impressive tour de force study that will be a major landmark in the field. I commend the authors for their truly impressive efforts and results.

Please address:

1. Clarify to the naïve reader that the over-synthesis / accumulation of LPS and the under-production of LPS are both lethal conditions for *E. coli*, but for different reasons / physiological consequences.
2. The authors repeatedly indicate that genetic studies have implicating the role of YejM and LapB in LpxC regulation through FtsH, but this ignores recent AP-MS studies that have demonstrated a more direct link (e.g. ref 29, Clairfeuille et al). For example, when describing YejM “All studies leading to this model were carried out using genetic approaches, however, leaving the molecular mechanisms of regulation unresolved”; this statement does not accurately reflect the recent literature, so please revise the text appropriately.
3. The recent models of YejM functioning directly as a periplasmic LPS sensor are not appropriately introduced in the introduction, and it would be best if the authors provided a more complete view of the state of the field and models before launching into their study results.
4. “However, co-expressing and purifying the His-tagged LapB chimeras with FtsH resulted in significantly reduced levels of FtsH co-elution (Fig. 1b)”. How are the authors quantifying this? Not all of these IPs look so significantly impacted. Are these chimeras functionally impaired in cells or not?
5. Can the authors state (in the results) what the detergent condition of their structure was determined. And if lipids added back through purification as well? Btw, how can you be certain the detergent condition is not artifactually impacting the oligomeric state? Can the authors rule out that the dimer-of-dimers arrangement is not a detergent artifact?
6. How did the authors assign phosphatidic acid? To my knowledge, this lipid is not appreciably present in *E. coli* (e.g. PMID: 25862689).

7. Can the authors state and provide a comparison to how the periplasmic domain of YejM in their structure compares to the EptA crystal structure, since dynamics of the domain in EptA had been previously discussed (PMID: 28193899)

8. Are any of the YejM or LapB residues that contact lipids in the autoinhibited structure essential; i.e. does their mutation impact regulation in cells.

9. It seems incredible that the amphipathic lipid A coordinating region of YejM appears “lifted” out of the membrane plane in the authors LapB complex. Can the authors highlight this point, since it also suggests there must be some intrinsic driving force for this region to repartition into the membrane (and presumably disrupt the LapB-dimer of dimers structure).

10. “In contrast to a previous report inferring that YejM and LapB form a constitutive complex²⁹, comparison of our YejM/LapB structure with the YejM/LPS crystal structure revealed that binding of LPS or LapB to YejM is mutually exclusive.” While it is very reasonable that LPS and LapB binding to YejM must be mutually exclusive, the authors have not firmly demonstrated that LapB and YejM completely disassociate when LPS becomes bound to YejM and LapB engages FtsH. So the possibility of a constitutive YejM/LapB complex is not necessarily ruled out, but instead a dynamic complex suggested implied. The authors should clarify there is more than one possibility of finite details here, and that their data or model do not absolutely resolve all of these.

General comment:

Structural figures are either small in rendering (relative to table or graph data) and can be improved. Fig 2c is quite small in scale, please consider to make this bigger. It’s unclear exactly what molecular interactions are being shown between residues using the dashed lines in Fig3b, please define better. In general, the quality of the structure figure rendering can be improved to provide greater clarity; consider using different shades of colors, or lighten the cartoon color when highlighting side-chains; also consider to use shadowing to provide better context for depth (e.g. Fig3c)

Reviewer #3 (Remarks to the Author):

Shu and Wi report a biochemical and structural analysis of the two essential membrane proteins, LapB (aka YciM) and YejM, that control the degradation of LpxC by the AAA+ protease FtsH. Based on the data obtained, they propose a model (Figure 5) showing how these regulatory proteins coordinate the synthesis and assembly of lipopolysaccharide (LPS) in the outer membrane of the Gram-negative bacterium *Escherichia coli*.

Based on genetic data, it had long been assumed that LapB was an adaptor protein that specifically directed LpxC to the FtsH. In Figure 1, they describe an in vitro assay using purified components that demonstrates this directly. LapB increases the K_M of LpxC binding to the protease, but it does not increase the rate of degradation. This is well done and is an important contribution.

The structures of LapB and YejM were already known, but here they use Cryo-EM to determine the structure of the YejM-LapB complex (Figure 2). The resolution is not great, the periplasmic domain of YejM is not visible, and parts of what they call LapB_B are missing. However, it appears that a dimer of the YejM membrane domains traps a dimer of the single trans-membrane helix (TMH) of LapB at the subunit interface.

A more detailed view of this LapB-YejM interaction is shown in Figure 3. It appears that the LapB TMH interacts with one YejM subunit in the cytoplasmic leaflet of the inner membrane and with the other in the periplasmic leaflet. There may be phospholipids in the empty spaces created in each leaflet by this unequal interaction, but again, the resolution does not allow identification of the molecules involved. As noted in the Discussion, this novel interaction may prevent high affinity association allowing a easily reversible interaction. It could also indicate a additional function for YejM. However, both of these proposals remains speculative as no experimental evidence is presented to support these predictions.

When comparing their YejM structure to the published YejM crystal structure, they notice a significant movement of the so-called linker region that has been shown to be important for LPS binding (Figure 4). Based on this observation they propose that YejM binding to LPS or to LapB are mutually exclusive, and this forms the basis for their model (Figure 5). While they do show that LPS reduces LpxC degradation in vitro, this result per se, sheds no direct light on the mechanism by which this occurs.

Their model shown in Figure 5 is appealing in its simplicity. However, as noted above, no experimental evidence is presented to support it and this is needed because two different laboratories have published in vivo data showing that a YejM-LapB interaction remains even when the linker region is removed. They note one of these publications (Ref 29), but not the other (Ref 25), and they make no attempt to address this direct challenge to their model.

Reviewer #4 (Remarks to the Author):

The manuscript by Shu and Mi covers an interesting and developing topic on the regulation of lipopolysaccharide (LPS) biosynthesis, particularly at its first committed step of biosynthesis that is mediated by LpxC. LpxC is an unstable protein and bacteria adjust its levels as per the demand for the LPS synthesis, which is partly dictated by cellular growth rate. Recently, an additional essential player YejM (LapC) that controls LpxC degradation, by modulating FtsH-LapB activity acting in an antagonistic manner to LapB, was described. In the present study, the authors further elaborate on this antagonistic action of LapC on LapB and how LapB enhances the FtsH-mediated proteolysis of LpxC but not of other FtsH substrates that were tested. These are the main highlights of this manuscript. This study is important and interesting, however certain additional information/experiments are required to substantiate these findings as described below.

1. Introduction section: LpxC is a known substrate for the FtsH protease, which requires LapB protein for proteolytic control of LpxC. LapB was already shown to form a complex with FtsH in 2014 and was also found to interact with LPS biosynthetic enzymes and LPS transport proteins (ref. 23). The authors should mention it in the introduction. The function of LapB, besides participating in regulating the FtsH proteolytic activity towards LpxC, needs to be emphasized in the introduction section. The authors should also mention proteolytic control of LpxC mediated by HslVU in the introduction section (ref. 26). LapC (YejM) was found to act as an antagonist of LapB, since suppressors mapping to the lapC gene can allow a deletion of the lapB gene (ref. 26), and LapC and LapB were found to co-purify (ref. 26 and 29). The authors do not mention discovery of genetic isolation of suppressors of a lapB deletion mapping to the lapC gene and biochemical/biophysical evidence of restoration of LPS content based on analysis of LPS composition by mass spectrometry. Thus, it is incorrect to state "All studies were". Please modify the sentence.
2. As the function of YejM as an interacting partner of LapB is now well established even before the submission of this manuscript, it is better to use LapC nomenclature rather than Y terminology when a function is unknown.
3. Lines 45-46 "remarkably..... The sentence needs to be modified as in the absence of LapB the activity of various stress response regulators such as RpoH, Cpx and RpoE is elevated. Ref. 21 study has examined the level of RpoH with just one condition and is not a detailed study.
4. Line 107 "Genetic evidence ... and line 108to form complex". Ref. 26 should be added as a complex is shown in that study as well based on co-purification and also at the genetic level.
5. Section YejM senses..... . It needs additional studies such as measurement of LPS levels at different growth stages when cell generation time is different, which is particularly relevant concerning the discussion section later on line 227.
6. Section YejM senses..... The authors need to provide a titration experiment with Kdo2-lipid A and that data should be presented.
7. Section LapB and LPS bind..... The authors must construct some mutants in LapB and YejM in the LPS-binding domain that disrupt the interaction and examine them for LPS and LpxC degradation. Authors need to demonstrate that residues 210-217 do not constitute the LPS-binding site in YejM.

8. Lipid-mediated interaction section line 146 and description in this section. The authors must identify which phospholipid is present in the YejM-phospholipid complex. Model has to be supported by experimental identification of phospholipid.

9. Discussion section line 187 providing another explanation for LapB nomenclature. The authors have not examined any other properties and functions of LapB. *lapB* mutants have complex phenotypes. In the absence of LapB, besides alteration of LpxC stability, such bacteria have LPS biosynthetic defects as observed by the accumulation of LPS precursor species, such bacteria also show the presence of significant amounts of aggregation-prone species of LpxM and other biosynthetic enzymes (ref. 23). Thus, LapB nomenclature for “LPS assembly protein” still is more appropriate. The authors should modify these sentences.

10. Discussion section. Lines 221 onwards about the regulation of LPS levels. A similar model has been proposed earlier (Int. J. Mol. Sci. 2022, 23(1), 189) and should be cited.

Minor comments:

(a) Ref. 22 is irrelevant as the *yciM* gene was identified even before that study with cell envelope defects, a role in thermotolerance, antibiotic sensitivity, biofilm formation, etc. Ref. 22 has not even one mention of LPS and in that study, this gene was not even found to be essential. This reference should be removed.

(b) Line 194. Replace *YciM* by LapB.

Referees' comments:

Reviewer #1 (Remarks to the Author):

The manuscript by Shu and Mi described the cryo-EM structure of YejM/LapB complex and its regulatory role in LPS synthesis through in vitro biochemical assays. The authors found that LapB is an adapter protein for FtsH mediated degradation of LpxC, an essential enzyme in LPS biosynthesis. They further discovered that a linker region of YejM, which had been found to bind LPS in a previous crystal structure, changed the conformation upon binding to LapB, which would not allow LPS binding. Thus, the authors proposed that YejM could sense the LPS concentration in the inner membrane and sequester LapB (forming the complex) when LPS concentration is low, which then prevents the degradation of LpxC and encourages the biosynthesis of LPS (a feedback loop). LPS is involved in the immune response of many pathogenic gram-negative bacteria. Therefore, it is very important to elucidate the structural and regulatory mechanism involved in the biosynthesis of LPS. This paper revealed the molecular basis underlying one of the regulatory circuits discovered from genetic studies. The assays developed in this work could be used for future studies in the field.

There are no outstanding concerns. I will support publication if the authors could address my questions and comments:

1. Does LapB only interact with FtsH through the TM domains? Are there any binding between the cytosolic domains of LapB and FtsH? It seems from affinity purification in Fig. 1b that other transmembrane helices (AcrZ, KdtA) also interact with FtsH although with weaker affinities. Are there any similarities in the TM sequences?

LapB interacts with FtsH through both the TM and cytoplasmic domains. We recently overexpressed the cytoplasmic domains of LapB. Although most overexpressed LapB_{cyto} are in the cytoplasm, a small fraction of LapB_{cyto} is attached to membrane fractions. We further purified the overexpressed LapB_{cyto} and observed that FtsH co-eluted with LapB_{cyto}, albeit in a very small amount (suggesting the TM helix contributes majorly to the interactions). Thus, in addition to TM domains, the cytoplasmic domains of LapB and FtsH also contribute to their interactions, which explains that LapB chimeras with the TM helix replaced still have weak affinities to FtsH.

There are no significant sequence similarities in the TM sequences among LapB, AcrZ, KdtA, and DjIA.

2. What is the stoichiometry between FtsH and LapB? Could it be measured? From the cryo-EM structure, LapB forms a dimer when bound to YejM. Does LapB also form a dimer when functioning as an adaptor?

The stoichiometry between FtsH and LapB is unknown. Our preliminary data (not shown in the manuscript) suggest that LapB is still a dimer when binding to FtsH.

3. Why FtsH did not degrade RpoH, CII, and Casein in Extended Data Fig. 2 (minimal activity was observed for LpxC but not the others in the absence of LapB)? A positive control would be better. Our results clearly demonstrated that FtsH did degrade β -casein efficiently (Extended data Fig. 2, more than 80% of β -casein were degraded in the lane of FtsH with ATP). Therefore, β -casein has already been served as a positive control to demonstrate that FtsH alone has strong protease activity. We speculate that the susceptibility of β -casein to FtsH is because of the low-energy barrier to unfold β -casein (Of 209 amino acid residues of β -casein, 16.7% are prolines, which evenly distribute along the sequence and limit the formation of α -helices). In contrast, RpoH and CII were degraded by FtsH alone slowly, perhaps because of no specific adaptors around for RpoH and CII.

4. In the LpxC degradation assay (Extended Data Fig. 3a), does TCA also precipitate part of the digested peptides? Would a non-specific protease (such as proteinase K) serve as a control to calibrate the effect of TCA?

Point well taken. Time-dependent release of TCA soluble peptides has been a widely accepted method to measure the protease activity of AAA+ proteases, including ClpXP and ClxAP (S Gottesman *et al.*, *Genes Dev* . 1998 May 1;12(9):1338-47) and FtsH (Herman C *et al.* *Mol Cell* . 2003 Mar;11(3):659-69) for more than two decades. Especially, combining fluorescence-labeled substrates (β -casein) and TCA precipitation to measure FtsH protease activity has been reported by several groups, including Bieniossek C *Proc Natl Acad Sci U S A* . 2009 Dec 22;106(51):21579-84. and Prabudiansyah I *Biochim Biophys Acta Biomembr* . 2021 Feb 1;1863(2):183526. Given that TCA precipitation has been a standard method to separate the digested peptides and undigested protein substrates for AAA+ protease assays, we had not considered that TCA could precipitate part of peptides, although we were the first group to apply this method to LpxC.

However, the reviewer raised an important question that we really like. As the Atto488-labeled LpxC allows us to visualize each fraction by in-gel fluorescence, we performed the following experiments: after digestion of fluorescent-labeled LpxC, we run SDS-PAGE for each fraction (digested samples before adding TCA, fractions of supernatant and precipitate after incubating with TCA and centrifugation) and use in-gel fluorescence to detect the digested products. In-gel fluorescence showed that the digested products were peptides with a range of molecular weights (smear bands). We did observe that peptides with relatively large molecular weights were precipitated by TCA, but the precipitated peptides were a small fraction of the total peptide products. Therefore, TCA precipitation did introduce a systematic error in the assay, and in turn, the proteases activity of FtsH was underestimated with the TCA precipitation approach (this likely happened in all literature that used the TCA method, including publications listed above). However, this error cannot be easily calibrated by using a non-specific protease (such as protease K). It is difficult to control proteinase K digesting LpxC to generate product peptides with a similar molecular weight range as FtsH-mediated proteolysis. Furthermore, our purpose was to compare the protease activity of FtsH and the FtsH/LapB complex. Considering that the underestimated protease activity was equally applied to FtsH alone and the FtsH/LapB complex., our conclusion that LapB improved the affinity but not V_{max} for LpxC degradation is not affected by this error. Therefore, we still present our results without further calibration. We appreciate that reviewer's question helped us to identify a potential weakness in our assay.

5. How does LapB interact with LpxC? Why LapB_{cyto} could inhibit the protease activity? Does that mean the interactions through TMs of LapB and FtsH are necessary for the activity? The authors should provide some explanations/discussions. This is related to question 1 and 2.

LapB uses its cytoplasmic domain to interact with LpxC (we added a sentence comparing K_d and K_m in the Results of the revised manuscript: “This K_d is similar to the K_M value (1.93 μM) in the kinetic studies of LpxC degradation by the FtsH/LapB proteoliposomes, which suggests binding between LpxC and the cytoplasmic domain of LapB is a major contributor to the increased affinity of LpxC to the FtsH/LapB complex. ”). Please refer to our Extended data Fig 3d to understand how LapB_{cyto} inhibits LpxC degradation: in our *in vitro* assay, full-length LapB and FtsH were reconstituted in proteoliposomes, the LapB/FtsH complex in proteoliposomes digested LpxC in solution (Extended data Fig 3d left). When we added the purified LapB_{cyto} into the solution, a truncated soluble form of LapB without TM helix should function as a ‘decoy’ adaptor, sequestering LpxC in solution and preventing LpxC from binding to the full-length LapB in the FtsH/LapB complex in the proteoliposomes and, therefore, LpxC degradation by the FtsH/LapB proteoliposomes is inhibited (Extended data Fig 3d right). We have revised the language related to this experiment in the Results and have added more explanation in the figure legend of Extended Data Fig 3 to make this easily understood.

6. Some 2D class averages showed both copies of YejM periplasmic domain (row1, column3 and row2, column3), why they were missed in the 3D reconstruction? Did the authors try focused classification with masks around the periplasmic domains?

2D class averages are from 2D images, which are projections of a 3D molecule. As long as the periplasmic domains move in a certain direction, projections along that direction should be clearly shown in the 2D class averages. However, in this case, the periplasmic domain will still be averaged out in 3D reconstruction because of moving in one direction. This might explain why we could see periplasmic domains in some 2D averages but cannot determine the high-resolution structure of this region in the 3D map. We had tried various focused classifications, but after extensive testing, we still could not significantly improve the resolution of the periplasmic domains.

7. Did the authors try mutating the LPS binding linker in YejM? Does that affect the binding of LapB? We performed a new experiment, introducing mutation in the LPS-binding linker region of YejM (YejM^{T213DR215DR216D}) and carried out *in vitro* assays with this mutant. New results have been included in our revised manuscript (Fig. 4d). In our assay, the YejM mutant still inhibits LpxC degradation as the wildtype YejM, suggesting that the mutations in the linker region do not affect YejM’s anti-adaptor function, i.e., the YejM mutant still binds to LapB as the wildtype YejM. This result is consistent with our model and the YejM/LapB complex structure, which clearly demonstrates that interactions between YejM and LapB are through the TM domains, not involving the linker region. Therefore, mutations in the linker region have no effect on YejM and LapB binding.

8. The statistics of cryo-EM model looks fine. A model vs. map FSC is missing. Also it is necessary to show the threshold/contour level/rmsd for all the maps displayed.

FSC between the PDB model and map has been added in the revised manuscript (Extended Figure 6c). Thresholds are added in the figure legends as well.

9. Some words were misused.

e.g., in the abstract "...demonstrating that lapB is an adapter protein, which explores its transmembrane helix to interact with..."

Page 4, last paragraph, "Since LapB exploits its TM helix to interact with..."

Thanks for careful reading, and words have been corrected as suggested.

Reviewer #2 (Remarks to the Author):

- >very noteworthy results
- >work will be a huge impact on the field and significantly advance the understanding of LPS synthesis/regulation and feedback
- >conclusions and claims are generally well supported by the data
- >some revision / clarification is needed, as exemplified below
- >solid methodology and standards met
- >sufficient detail in methods provided

Multi-drug resistant Gram-negative bacterial (GNB) pathogens represent an urgent global threat. The outer membrane (OM) of GNB is essential for life and virulence and forms a major barrier to most antibiotic discovery efforts. The unique lipopolysaccharide (LPS) molecule found in most GNB is a key molecule within the OM, and its synthesis is tightly regulated through bacterial growth, but the underlying mechanisms of this regulation have remained enigmatic for >30 years. The present study by Shu and Mi sheds key and new light on the molecular and structural details of LPS regulation, and is truly an impressive tour de force study that will be a major landmark in the field. I commend the authors for their truly impressive efforts and results.

Please address:

1. Clarify to the naïve reader that the over-synthesis / accumulation of LPS and the under-production of LPS are both lethal conditions for *E. coli*, but for different reasons / physiological consequences.

We revised the first paragraph of the Introduction to have these included: "As a lipid essential for the viability of most Gram-negative bacteria, LPS synthesis is under tight control: too little LPS compromises the outer membrane (OM), triggering cell envelope stress responses and leading to cell death; too much LPS breaks the balance between LPS and phospholipids, and accumulation of LPS in the IM is toxic and lethal. In *E. coli*, cellular levels of LPS are controlled by LpxC, a deacetylase that performs the first committed step of synthesizing LPS."

2. The authors repeatedly indicate that genetic studies have implicating the role of YejM and LapB in LpxC regulation through FtsH, but this ignores recent AP-MS studies that have demonstrated a more direct link (e.g. ref 29, Clairfeuille et al). For example, when describing YejM "All studies leading to this model were carried out using genetic approaches, however, leaving the molecular mechanisms of regulation unresolved"; this statement does not accurately reflect the recent literature, so please revise the text appropriately.

We appreciate that the reviewer pointed this out. We have revised the second paragraph in the Introduction to properly reflect the current progress in the field.

3. The recent models of YejM functioning directly as a periplasmic LPS sensor are not appropriately introduced in the introduction, and it would be best if the authors provided a more complete view of the state of the field and models before launching into their study results.

Points are taken. We have emphasized the importance of the YejM/LPS structure in the Introduction: "Intriguingly, an LPS molecule was found binding to YejM in a recent crystal structure of YejM,

suggesting that YejM is a sensor of LPS. This discovery provides the first glimpse of how LPS levels are sensed in the IM to regulate LpxC degradation.”

4. “However, co-expressing and purifying the His-tagged LapB chimeras with FtsH resulted in significantly reduced levels of FtsH co-elution (Fig. 1b)”. How are the authors quantifying this? Not all of these IPs look so significantly impacted. Are these chimeras functionally impaired in cells or not?

The densities of the bands of co-eluted FtsH in SDS-PAGE with different LapB chimeras were quantified as numbers below each lane of elution (Fig. 1b). We tested the stimulatory effect of these chimeras, and the stimulation of LpxC degradation correlated well with the binding affinity between the LapB chimeras with FtsH (data not shown in the manuscript). The approaches in our lab are mainly biochemistry and structural analysis, and a platform for editing bacterial genomic DNA and testing the phenotype has not been established in our lab; therefore, we cannot assess the effects of these chimeras *in vivo*.

5. Can the authors state (in the results) what the detergent condition of their structure was determined. And if lipids added back through purification as well? Btw, how can you be certain the detergent condition is not artifactually impacting the oligomeric state? Can the authors rule out that the dimer-of-dimers arrangement is not a detergent artifact?

We used the detergent GDN for structure determination and stated this in the Result part of the revised manuscript. We did not add any lipid during purification. Therefore, the lipid molecules observed at the YejM and LapB interfaces were endogenously co-purified. To excluded the artifact caused by detergent, we have purified the YejM/LapB complex with styrene maleic acid (SMA). Negative stain EM images and 2D averages of SMA purified YejM/LapB showed a similar oligomeric state of the YejM/LapB complex in GDN (Extended Fig. 4d-f). Therefore, our new experiment results strongly support that our structure represents a native state.

6. How did the authors assign phosphatidic acid? To my knowledge, this lipid is not appreciably present in *E. coli* (e.g. PMID: 25862689).

In our cryoEM map, there are no densities for a large head group from the lipid_{cyto}. Therefore, we speculate it is either diacylglycerol (DAG) or phosphatidic acid (PA). As Arg22 is close to the lipid head, it is more reasonable to model PA, as the phosphate group of PA perfectly interact with Arg22. In the Discussion, we postulate that YejM is not only a sensor for LPS but may also sense the level of phospholipids in the inner membrane, thus balancing LPS and phospholipids synthesis. PA is a common precursor for all glycerophospholipids (PMID: 2404013), and their low percentage in the IM allows the fluctuations of PA concentration more easily to be detected. Therefore, we think PA is a perfect signal molecule to reflect the status of phospholipid synthesis. We have included this explanation in the revised Discussion. We are collaborating with a native MS group to identify the lipids in the YejM/LapB complex. However, because of some technical challenges, we cannot define the lipid identity in a short time. For more details, please also see our answer to question 8 from reviewer 4.

7. Can the authors state and provide a comparison to how the periplasmic domain of YejM in their structure compares to the EptA crystal structure, since dynamics of the domain in EptA had been

previously discussed (PMID: 28193899)

We included a structural comparison of YejM and EptA (Extended Data Fig. 8f). We also have added a new paragraph in the Discussion to explain what role the dynamics of YejM's periplasmic domain may play in its function.

8. Are any of the YejM or LapB residues that contact lipids in the autoinhibited structure essential; i.e., does their mutation impact regulation in cells.

We did mutate Arg22 in LapB, which is supposed to interact with the head group in the lipid_{cyto} (we model PA as lipid_{cyto}), but mutants (R22A or R22D) had no effect in our biochemical assays of LpxC degradation (not shown in the manuscript). Considering the extensive interactions between the acyl chains of lipids with YejM/LapB and inaccurate modeling of the acyl chains (the exact position of each carbon atom in the acyl chain cannot be determined merely based on the density map), we cannot accurately identify what residues in YejM and LapB interact with the acyl chains. As we explained in our answer to question #4 above, we are not able to introduce mutations in *E. coli* genome, therefore cannot evaluate how mutations impact regulations in cells (we do not expect R22A or R22D will have any effect *in vivo*, given they have no effects with our *in vitro* assays).

9. It seems incredible that the amphipathic lipid A coordinating region of YejM appears “lifted” out of the membrane plane in the authors LapB complex. Can the authors highlight this point, since it also suggests there must be some intrinsic driving force for this region to repartition into the membrane (and presumably disrupt the LapB-dimer of dimers structure).

We appreciate the reviewer bringing up this important point, which we did not think through before. As suggested, we have a new paragraph in Discussion, which explain how we think about this and how the conformational changes in the linker region may relate to the flexibility of the periplasmic domain.

10. “In contrast to a previous report inferring that YejM and LapB form a constitutive complex²⁹, comparison of our YejM/LapB structure with the YejM/LPS crystal structure revealed that binding of LPS or LapB to YejM is mutually exclusive.” While it is very reasonable that LPS and LapB binding to YejM must be mutually exclusive, the authors have not firmly demonstrated that LapB and YejM completely disassociate when LPS becomes bound to YejM and LapB engages FtsH. So the possibility of a constitutive YejM/LapB complex is not necessarily ruled out, but instead a dynamic complex suggested implied. The authors should clarify there is more than one possibility of finite details here, and that their data or model do not absolutely resolve all of these.

Clairfeuille et al concluded that YejM and LapB form a constitutive complex based on two-hybrid experiments. However, we hope to point out an inherent caveat in drawing such a conclusion from their experiments. In their two-hybrid experiments, as long as YejM and LapB interact during a certain phase of cell growth, such as the log phase, the YejM/LapB complex will bring T25 and T18 together, catalyzing the reaction and turning the color of colonies to blue. Even later in the stationary phase, the YejM/LapB dissociate because of a high LPS level in the IM, the activated adenylate cyclase has catalyzed the reaction, and the blue colonies will not be back to white. Therefore, the two-hybrid experiments by Clairfeuille *et al.* only support that YejM and LapB form a complex but cannot tell whether it is a transient or constitutive complex. We pointed this out in our Discussion.

Our YejM/LapB complex structure perfectly explains the two-hybrid experiment results carried out by Clairfeuille: the interactions between YejM and LapB are through their TM domains in our structure, and the two-hybrid experiments also show that the TM domain in YejM and LapB have blue colonies. We also performed more experiments, showing that incubation of purified YejM/LapB complex with LPS caused the complex dissociation (Extended Data Fig. 11). Although the purified YejM/LapB in detergent micelles and incubation with LPS cannot fully represent their behavior in the native membrane environment, our new biochemical data support our claim that bindings of LapB or LPS are mutually exclusive. Following the reviewer's suggestion, we mentioned in the revised manuscript that whether the complex is constitutive requires more *in vivo* studies.

General comment:

Structural figures are either small in rendering (relative to table or graph data) and can be improved. Fig 2c is quite small in scale, please consider to make this bigger. It's unclear exactly what molecular interactions are being shown between residues using the dashed lines in Fig3b, please define better. In general, the quality of the structure figure rendering can be improved to provide greater clarity; consider using different shades of colors, or lighten the cartoon color when highlighting side-chains; also consider to use shadowing to provide better context for depth (e.g. Fig3c)

To address these concerns, we have remade structural figures by ChimeraX.

Reviewer #3 (Remarks to the Author):

Shu and Wi report a biochemical and structural analysis of the two essential membrane proteins, LapB (aka YciM) and YejM, that control the degradation of LpxC by the AAA+ protease FtsH. Based on the data obtained, they propose a model (Figure 5) showing how these regulatory proteins coordinate the synthesis and assembly of lipopolysaccharide (LPS) in the outer membrane of the Gram-negative bacterium *Escherichia coli*.

Based on genetic data, it had long been assumed that LapB was an adaptor protein that specifically directed LpxC to the FtsH. In Figure 1, they describe an in vitro assay using purified components that demonstrates this directly. LapB increases the K_M of LpxC binding to the protease, but it does not increase the rate of degradation. This is well done and is an important contribution.

We appreciate this comment.

The structures of LapB and YejM were already known, but here they use Cryo-EM to determine the structure of the YejM-LapB complex (Figure 2). The resolution is not great, the periplasmic domain of YejM is not visible, and parts of what they call LapB_B are missing. However, it appears that a dimer of the YejM membrane domains traps a dimer of the single trans-membrane helix (TMH) of LapB at the subunit interface.

A more detailed view of this LapB-YejM interaction is shown in Figure 3. It appears that the LapB TMH interacts with one YejM subunit in the cytoplasmic leaflet of the inner membrane and with the other in the periplasmic leaflet. There may be phospholipids in the empty spaces created in each leaflet by this unequal interaction, but again, the resolution does not allow identification of the molecules involved. As noted in the discussion, this novel interaction may prevent high affinity association allowing a easily reversible interaction. It could also indicate a additional function for YejM. However, both of these proposals remains speculative as no experimental evidence is presented to support these predictions.

The TM regions in our structure have a resolution of $\sim 3 \text{ \AA}$, which is high enough to show most sidechains of TMH and densities of phospholipid molecules. This allows us to confidently claim two phospholipid molecules at the interface between YejM and LapB, although the structural information is not enough for us to define what the two phospholipid molecules are. We hope to point out that identifying lipids molecules in membrane protein structures is a very challenging task due to the limitation of techniques (a recent review PMID: 33930613). Even if we could further improve the resolution to atomic resolution (this is very difficult!), unambiguous assignment of lipids still requires other techniques, such as native mass spectrometry. An example: an atomic resolution (1.8 \AA) structure (PDB 4UC1) could not determine what lipid was in the structure, and the final identification required native mass spectrometry (please also see our answer to question 8 from reviewer 4). We are collaborating with a native mass spectrometry lab and trying to identify the lipids. However, there are some technical challenges that we cannot overcome in a short time. We want to emphasize that previous crystal structures of YejM also observed phospholipids at the same position in our YejM/LapB complex structure. However, researchers in two groups have overlooked the potential importance of these phospholipids because they only had the YejM structure and there was no way for them to predict the lipids mediating

interactions with LapB only based on the YejM crystal structure. Our finding of lipids at the YejM and LapB interfaces reveals the potential role of YejM: not only sensing LPS but also may serve as a phospholipid sensor. We hope this prediction of YejM's new function will pique more interest in this overlooked direction.

When comparing their YejM structure to the published YejM crystal structure, they notice a significant movement of the so-called linker region that has been shown to be important for LPS binding (Figure 4). Based on this observation they propose that YejM binding to LPS or to LapB are mutually exclusive, and this forms the basis for their model (Figure 5). While they do show that LPS reduces LpxC degradation *in vitro*, this result *per se*, sheds no direct light on the mechanism by which this occurs.

We appreciate that the reviewer noticed the conformational changes in the linker region. However, the reviewer missed one important piece of information in our manuscript: LPS and LapB dimer have overlapping binding sites in YejM (Fig. 4 b&c), which is the basis of our claim that YejM binding to LPS or to LapB is mutually exclusive. To present the overlapping binding sites of LPS and LapB more clearly, we have added a movie to the revised manuscript (Extended Data Movie 1). Furthermore, we carried out more biochemical experiments to support this claim: we incubated the purified YejM/LapB complex with LPS and found the complex dissociated, and in controls with cardiolipin or POPC, the complex was stable (Extended Data Fig. 11). Although incubation of the purified YejM/LapB with LPS cannot fully represent their behavior in a native membrane environment, our new biochemical data support our claim that the binding of LapB and LPS to YejM is mutually exclusive.

Their model shown in Figure 5 is appealing in its simplicity. However, as noted above, no experimental evidence is presented to support it and this is needed because two different laboratories have published *in vivo* data showing that a YejM-LapB interaction remains even when the linker region is removed. They note one of these publications (Ref 29), but not the other (Ref 25), and they make no attempt to address this direct challenge to their model.

Our structure clearly demonstrates that interactions between YejM and LapB are completely through their TM helices and that the linker region is not involved in YejM/LapB interactions. Therefore, our YejM/LapB complex structure perfectly explains the genetic results that deletion of the periplasmic domain and the linker in YejM still allows forming a complex with LapB. We do not see any contradiction between our structure and the genetic results related to YejM truncations.

Based on the two-hybrid experiments, Clairfeuille et al. claimed that YejM and LapB form a "constitutive" complex. It is improper to make such a conclusion based on bacterial two-hybrid assays: if YejM and LapB form a transient complex, the colonies will turn blue, and after the complex dissociates, the blue colony will not turn back to white as the reaction has been catalyzed. Therefore, from the bacterial two-hybrid assays, Clairfeuille et al. should reach a conclusion that YejM and LapB can form a complex, but whether the complex is constitutive or transient requires other experiments to test. Fivenson and Bernhardt used a POLAR two-hybrid assay to prove YejM and LapB interactions, but they did not claim that YejM and LapB form a constitutive complex. Our structural and biochemical results suggest that the complex of YejM/LapB is reversible and that formation and dissociation of the

YejM/LapB complex are central to the regulation of LpxC degradation. We include a paragraph in the Discussion of the revised manuscript to address this issue.

We agree that the regulation of LPS synthesis is very likely more complicated than the model we proposed here. We pointed out that our model is a “simplified” model in the last sentence in the revised manuscript.

Reviewer #4 (Remarks to the Author):

The manuscript by Shu and Mi covers an interesting and developing topic on the regulation of lipopolysaccharide (LPS) biosynthesis, particularly at its first committed step of biosynthesis that is mediated by LpxC. LpxC is an unstable protein and bacteria adjust its levels as per the demand for the LPS synthesis, which is partly dictated by cellular growth rate. Recently, an additional essential player YejM (LapC) that controls LpxC degradation, by modulating FtsH-LapB activity acting in an antagonistic manner to LapB, was described. In the present study, the authors further elaborate on this antagonistic action of LapC on LapB and how LapB enhances the FtsH-mediated proteolysis of LpxC but not of other FtsH substrates that were tested. These are the main highlights of this manuscript. This study is important and interesting, however certain additional information/experiments are required to substantiate these findings as described below.

1. Introduction section: LpxC is a known substrate for the FtsH protease, which requires LapB protein for proteolytic control of LpxC. LapB was already shown to form a complex with FtsH in 2014 and was also found to interact with LPS biosynthetic enzymes and LPS transport proteins (ref. 23). The authors should mention it in the introduction. The function of LapB, besides participating in regulating the FtsH proteolytic activity towards LpxC, needs to be emphasized in the introduction section. The authors should also mention proteolytic control of LpxC mediated by HslVU in the introduction section (ref. 26). LapC (YejM) was found to act as an antagonist of LapB, since suppressors mapping to the lapC gene can allow a deletion of the lapB gene (ref. 26), and LapC and LapB were found to co-purify (ref. 26 and 29). The authors do not mention discovery of genetic isolation of suppressors of a lapB deletion mapping to the lapC gene and biochemical/biophysical evidence of restoration of LPS content based on analysis of LPS composition by mass spectrometry. Thus, it is incorrect to state "All studies were". Please modify the sentence.

We have revised the manuscript and given a more comprehensive introduction. We have also cited the references according to requests from the reviewer.

2. As the function of YejM as an interacting partner of LapB is now well established even before the submission of this manuscript, it is better to use LapC nomenclature rather than Y terminology when a function is unknown.

Several names have been proposed for YejM, including LapC, PbgA, and ClxD. However, YejM is still the most commonly used name in literature. We had deposited the structure to PDB under the name of YejM. To be consistent with the PDB deposit and most literature, we prefer to use YejM in the manuscript.

3. Lines 45-46 "remarkably..... The sentence needs to be modified as in the absence of LapB the activity of various stress response regulators such as RpoH, Cpx and RpoE is elevated. Ref. 21 study has examined the level of RpoH with just one condition and is not a detailed study.

Reference is added as suggested.

4. Line 107 “Genetic evidence ... and line 108to form complex”. Ref. 26 should be added as a complex is shown in that study as well based on co-purification and also at the genetic level.

Added as suggested.

5. Section YejM senses..... . It needs additional studies such as measurement of LPS levels at different growth stages when cell generation time is different, which is particularly relevant concerning the discussion section later on line 227.

We have isolated the IM and quantified LPS levels by western blot with LPS antibodies. New data has been included in the revised manuscript (Extended Data Fig. 16).

6. Section YejM senses..... The authors need to provide a titration experiment with Kdo2-lipid A and that data should be presented.

Extra experiments have been performed (Extended Data Fig. 12).

7. Section LapB and LPS bind..... The authors must construct some mutants in LapB and YejM in the LPS-binding domain that disrupt the interaction and examine them for LPS and LpxC degradation. Authors need to demonstrate that residues 210-217 do not constitute the LPS-binding site in YejM.

An extra YejM mutate has been constructed and tested in the *in vitro* assay (Fig. 4d). New results of the YejM mutant further support our model.

8. Lipid-mediated interaction section line 146 and description in this section. The authors must identify which phospholipid is present in the YejM-phospholipid complex. Model has to be supported by experimental identification of phospholipid.

Thanks for the suggestion. Identification of the phospholipid molecules at the interfaces between YejM and LapB in our PDB model is also our top priority. However, this is a challenging topic that cannot be simply made by conventional MS analysis of lipids, because conventional lipid MS can only identify lipids, but cannot distinguish the annular and specific lipid binding to the YejM/LapB complex (there are a lot of lipid molecules co-purified with proteins and surrounding the YejM/LapB complex). Identification of phospholipids can not be achieved by simply improving the resolution of the structure either. A recent example: an atomic-resolution (1.8 Å) structure that cannot determine what lipid is in the structure merely based on the density map of that lipid (PMID: **33930613**). To identify the lipid at the interface, we have to carry out native MS, see recent reviews (PMID: 31886601 and PMID: 33930613). We have reached out to a native MS lab (Dr. Kallol Gupta, <https://www.theguptalab.com/>) that has the expertise in identifying specific lipids in membrane proteins. We did some preliminary experiments. However, all efforts have failed because of the technical challenges in liberating the YejM/LapB complex from the detergent micelles. Our YejM/LapB structure was determined with detergent GDN, and GDN micelles cannot be removed by collisional activation while maintaining the lipid/protein interactions because of the tight binding of GDN to YejM/LapB. Our native MS collaborator

at Yale, Kallol Gupta, told us that his lab and Carol Robinson's lab (another world renowned native MS expert) had never successfully identified the lipid in membrane proteins purified with GDN. In summary, because of the current technical limits, we cannot identify the lipids in the structure of the YejM/LapB complex in the near future.

It is a common practice in the structural biology field to put some tentative models in the unassigned densities. We have claimed that our lipid models were tentative, and the modeling was based on the best information we could obtain from the structure. Furthermore, the identities of the phospholipids do not affect our hypothesis of how LPS regulates LpxC degradation, which is the major model proposed in the manuscript to explain the feedback control of LpxC degradation. We also pointed in the revised manuscript that to further understand how LPS and phospholipids synthesis is coupled, identifying the lipids at the YejM/LapB complex will be crucial.

9. Discussion section line 187 providing another explanation for LapB nomenclature. The authors have not examined any other properties and functions of LapB. lapB mutants have complex phenotypes. In the absence of LapB, besides alteration of LpxC stability, such bacteria have LPS biosynthetic defects as observed by the accumulation of LPS precursor species, such bacteria also show the presence of significant amounts of aggregation-prone species of LpxM and other biosynthetic enzymes (ref. 23). Thus, LapB nomenclature for "LPS assembly protein" still is more appropriate. The authors should modify these sentences.

We agree that LapB may have multiple functions in addition to the adaptor for LpxC degradation. However, we have a different opinion on whether LapB functions as LPS assembly protein. In the seminal JBC paper, Klein and colleagues showed complex phenotypes from LapB mutant. However, the authors did not further distinguish what defects are directly caused by the loss function of LapB and what defects are the downstream effects. In our opinion, the accumulation of LPS precursors and aggregation-prone species of LpxM are likely to be indirect results of the accumulation of LpxC in LapB mutants. This defect may result from the imbalance between the accumulated LpxC and enzymes that catalyze the late steps of LPS synthesis: the glycosyltransferases become limiting enzymes and cannot catalyze all precursors produced by accumulated LpxC, therefore accumulating a lot of LPS precursors. Aggregation of LpxM may be the result of a buildup of LPS in the inner membrane. In our lab, we have observed several proteins incubating with LPS resulted in migrating slower on SDS-PAGE (see the Extended Data 11, YejM and LapB bands also shifted in SDS-PAGE when LPS is present). Since Klein et al. observed a similar result in LpxM, we speculate that too much LPS accumulated in the IM (downstream effects from high levels of LpxC in the LapB mutant) slowed mobility of LpxM in SDS-PAGE. Klein et al. cited Ogura's work that FtsH mutants accumulate LpxC but had intact LPS to defend their claim of LapB functioning as an LPS assembly protein. However, FtsH has diverse substrates, and we don't think it is proper to compare the phenotypes of FtsH mutants and LapB mutants. In our opinion, proper control should be overexpression of LpxC to compare with the LapB mutants, such as evaluating LPS composition (whether LPS precursors accumulate) in the strains overexpressing LpxC. Unfortunately, Klein and her colleagues did not perform such control experiments, and we cannot tell whether those defect are just because of accumulation of LpxC.

Two excellent review papers (PMID: 32631947 by Joe Letkenhaus and PMID: 33036869 by Tom Silhavy) and a research paper from the Misra group (PMID: 32540932) still use the name of YciM, reflecting that

the experts in the field still do not fully accept the concept that LapB functions as an LPS assembly protein. We hope by providing an alternative explanation (LpxC degradation adapter protein B), LapB will become a more widely accepted name in the field.

10. Discussion section. Lines 221 onwards about the regulation of LPS levels. A similar model has been proposed earlier (Int. J. Mol. Sci. 2022, 23(1), 189) and should be cited.

Cited as suggested.

Minor comments:

(a) Ref. 22 is irrelevant as the yciM gene was identified even before that study with cell envelope defects, a role in thermotolerance, antibiotic sensitivity, biofilm formation, etc. Ref. 22 has not even one mention of LPS and in that study, this gene was not even found to be essential. This reference should be removed.

Points taken. We have deleted this reference from the Introduction.

(b) Line 194. Replace YciM by LapB.

Corrected

REVIEWERS' COMMENTS

Reviewer #1 (Remarks to the Author):

The authors have successfully addressed my concerns and greatly improved the clarity and depth of the paper. Therefore, I support the publication in Nature Communications. Here are some additional comments with no actions required from the authors.

1. Regarding the original comment 4, I encourage the authors to consider developing new assays in the near future to avoid the systematic error. It is true that the current assay has been used in previous works, but a better assay would benefit the field and may lead to new insights at the molecular level. This is really a good opportunity for a young investigator entering the field.
2. Regarding the original comment 5, I understand the assay. The question is more about the concept. The degradation of LpxC was inhibited in the presence of LapBcyto due to a competition of the substrates, does that equal to the statement that the protease activity of FtsH was inhibited? Careful wording is necessary in these statements.
3. Regarding the identification of the bound lipid raised by other reviewers, this is a really challenging task and is, in my opinion, beyond the scope of the paper. I'm fine with the current treatment, i.e., a putative phosphatidic acid was modeled and briefly discussed. The authors should also make it clear in the PDB depositions.

Reviewer #2 (Remarks to the Author):

Congratulations to the authors for improving their breakthrough study - this work significantly advances our understanding of this important physiology.

Please just note Clairfeuille et al did not perform any two-hybrid studies as the authors suggest, but instead IPs, MS and complementation studies.

Reviewer #3 (Remarks to the Author):

The authors did a good job of addressing most of this reviewer's concerns. They show that LPS dissociates the YejM/LapB interaction and that adding LPS to proteoliposomes containing the YejM LPS binding mutant does not increase LpxC degradation. However, the authors state that their model perfectly explains the genetic data when in fact, it does not. YejMT213D, which can no longer bind LPS, interacts with LapB and LpxC levels are increased. However, a mutant lacking the periplasmic and interfacial domains of YejM, which also presumably can no longer bind LPS, must still interact with LapB, but LPS levels are dramatically decreased. Indeed, two different labs have shown that truncated YejM still interacts with LapB. Thus, the authors proposed model is much too simple as it does not explain these facts.

Reviewer #4 (Remarks to the Author):

The revised manuscript by Shu and Mi is vastly improved. Authors have addressed most of the points raised by this referee. However, some changes are still required.

1. Line 36: The statement regarding "Transcription of the *lpxC* gene is stable" needs to be amended. Several microarray-based assays and transcriptome studies show the transcription of the *lpxC* gene is inducible and for example activated by SoxR/S system (PMID: 26279566) [Seo et al 2015].
2. Lines 300-301 regarding Extended data Fig. 16. It shows a massive retention of LPS in inner membranes in the stationary phase and that should virtually kill the bacteria. This figure should be removed as no control about fractionation of membrane proteins are shown to convince readers that inner membrane fractions are clean. Authors should note that the majority of structural work using purified LPS is obtained from bacteria when they are in the stationary phase and such LPS is often modified by nonstoichiometric incorporation of aminoarabinose and phosphoethanolamine (when modification systems are induced). Such modifications occur only on the periplasmic side, indicating transport of LPS across the IM.

Reviewer #1 (Remarks to the Author):

The authors have successfully addressed my concerns and greatly improved the clarity and depth of the paper. Therefore, I support the publication in Nature Communications. Here are some additional comments with no actions required from the authors.

1. Regarding the original comment 4, I encourage the authors to consider developing new assays in the near future to avoid the systematic error. It is true that the current assay has been used in previous works, but a better assay would benefit the field and may lead to new insights at the molecular level. This is really a good opportunity for a young investigator entering the field.

I very much appreciate this comment and we will consider developing better assays to overcome the systematic error introduced by TCA precipitation.

2. Regarding the original comment 5, I understand the assay. The question is more about the concept. The degradation of LpxC was inhibited in the presence of LapBcyto due to a competition of the substrates, does that equal to the statement that the protease activity of FtsH was inhibited? Careful wording is necessary in these statements.

Now, I fully catch the reviewer's point. I fully agree with the comment.

3. Regarding the identification of the bound lipid raised by other reviewers, this is a really challenging task and is, in my opinion, beyond the scope of the paper. I'm fine with the current treatment, i.e., a putative phosphatidic acid was modeled and briefly discussed. The authors should also make it clear in the PDB depositions.

Thanks for understanding the challenges. We will inform PDB that the lipid molecules in our model are tentative.

Reviewer #2 (Remarks to the Author):

Congratulations to the authors for improving their breakthrough study - this work significantly advances our understanding of this important physiology.

Please just note Clairfeuille et al did not perform any two-hybrid studies as the authors suggest, but instead IPs, MS and complementation studies.

We have revised our manuscript and deleted the portion commenting on the two-hybrid results in the Discussion.

Reviewer #3 (Remarks to the Author):

The authors did a good job of addressing most of this reviewer's concerns. They show that LPS dissociates the YejM/LapB interaction and that adding LPS to proteoliposomes containing the YejM LPS binding mutant does not increase LpxC degradation. However, the authors state that their model perfectly explains the genetic data when in fact, it does not. YejMT213D, which can no longer bind LPS, interacts with LapB and LpxC levels are increased. However, a mutant lacking the periplasmic and interfacial domains of YejM, which also presumably can no longer bind LPS, must still interact with LapB, but LPS levels are dramatically decreased. Indeed, two different labs have shown that truncated YejM still interacts with LapB. Thus, the authors proposed model is much too simple as it does not explain these facts.

We fully agree that our structural data cannot explain why deletion of the periplasmic domain reduces LPS levels. We have added one sentence in the Discussion: "With our current structural data, we also cannot explain why deletion of YejM's periplasmic domain causes reducing levels of LPS, and this effect may be the result of YejM antagonizing through approaches other than sequestering LapB from FtsH."

Reviewer #4 (Remarks to the Author):

The revised manuscript by Shu and Mi is vastly improved. Authors have addressed most of the points raised by this referee. However, some changes are still required.

1. Line 36: The statement regarding "Transcription of the *lpxC* gene is stable" needs to be amended. Several microarray-based assays and transcriptome studies show the transcription of the *lpxC* gene is inducible and for example activated by SoxR/S system (PMID: 26279566) [Seo et al 2015].

We appreciate the reviewer pointing this out. We have deleted the sentence in the revised manuscript according to the reviewer's suggestion.

2. Lines 300-301 regarding Extended data Fig. 16. It shows a massive retention of LPS in inner membranes in the stationary phase and that should virtually kill the bacteria. This figure should be removed as no control about fractionation of membrane proteins are shown to convince readers that inner membrane fractions are clean. Authors should note that the majority of structural work using purified LPS is obtained from bacteria when they are in the stationary phase and such LPS is often modified by nonstoichiometric incorporation of aminoarabinose and phosphoethanolamine (when modification systems are induced). Such modifications occur only on the periplasmic side, indicating transport of LPS across the IM.

Points well taken. We have removed the figure as required.